# Current Approaches for Combination Therapy of Cancer: The Role of Immunogenic Cell Death

**DOI:** 10.3390/cancers12041047

**Published:** 2020-04-23

**Authors:** Zahra Asadzadeh, Elham Safarzadeh, Sahar Safaei, Ali Baradaran, Ali Mohammadi, Khalil Hajiasgharzadeh, Afshin Derakhshani, Antonella Argentiero, Nicola Silvestris, Behzad Baradaran

**Affiliations:** 1Immunology Research Center, Tabriz University of Medical Sciences, Tabriz 5165665811, Iran; zahraasadzadeh2834@gmail.com (Z.A.); shar.safaee@gmail.com (S.S.); hajiasgharzadeh@yahoo.com (K.H.); afshin.derakhshani94@gmail.com (A.D.); 2Department of Immunology and Microbiology, Faculty of Medicine, Ardabil University of Medical Sciences, Ardabil 5618985991, Iran; safarzadehelham@yahoo.com; 3Research & Development Lab, BSD Robotics, 4500 Brisbane, Australia; ali_baradaran79@hotmail.com; 4Department of Cancer and Inflammation Research, Institute for Molecular Medicine, University of Southern Denmark, 5230 Odense, Denmark; ali_mohammadi6868@yahoo.com; 5IRCCS Istituto Tumori “Giovanni Paolo II” of Bari, 70124 Bari, Italy; argentieroantonella@gmail.com; 6Department of Biomedical Sciences and Human Oncology, University of Bari “Aldo Moro”, 70124 Bari, Italy; 7Department of Immunology, Faculty of Medicine, Tabriz University of Medical Sciences, Tabriz 5166614766, Iran

**Keywords:** immunogenic cell death, DAMPs, combination therapy

## Abstract

Cell death resistance is a key feature of tumor cells. One of the main anticancer therapies is increasing the susceptibility of cells to death. Cancer cells have developed a capability of tumor immune escape. Hence, restoring the immunogenicity of cancer cells can be suggested as an effective approach against cancer. Accumulating evidence proposes that several anticancer agents provoke the release of danger-associated molecular patterns (DAMPs) that are determinants of immunogenicity and stimulate immunogenic cell death (ICD). It has been suggested that ICD inducers are two different types according to their various activities. Here, we review the well-characterized DAMPs and focus on the different types of ICD inducers and recent combination therapies that can augment the immunogenicity of cancer cells.

## 1. Introduction

Immunotherapy has been considered as a promising therapeutic modality in oncology that aims to re-activate the immune system which is stopped by the tumor cells and creates a robust antitumor response [1]. Tumor cells, by having resistance to cell death and escaping from immunological surveillance, are durable in cancer patients [2]. So, restoring the susceptibility of cancer cells to death and intensifying immune recognition of poorly immunogenic tumor cells create strategies for therapeutic success [3]. The combination of these strategies, which would take advantage of the probable immunogenic traits of diverse forms of cancer cell death, is an interesting viewpoint [4]. The antitumor immune response can be provided by immunogenic cell death (ICD), a different class of cell death described by release or expression of calreticulin (CRT), adenosine triphosphate (ATP), high mobility group box 1 (HMGB1), heat-shock proteins (HSPs), ANXA1, and stimulation of type I interferon [5,6,7]. Binding of these danger-associated molecular patterns (DAMPs) to their receptors leads to immune cells’ recruitment and induction. Lastly, they lead to recognition, phagocytosis, and activation of T lymphocytes to eradicate cancer cells [8]. Up to now, some single-agent ICD inducers have been introduced, comprising standard chemotherapeutics, targeted anticancer factors, and numerous other treatment options [9,10]. It was documented that the immunogenicity of tumor cells can be promoted by particular treatments (e.g., chemotherapy, radiotherapy, and photodynamic therapy) and leads to antitumor immunity [11,12]. Here, we summarized the well-characterized DAMPs and discuss in detail the different inducers of ICD.

## 2. Immunogenic Cell Death: New Meaning in Cancer Therapy

Naturally, the immune system can recognize and destroy cancer cells and plays a significant role in the regulation of tumor progression. The immune system is educated in such a way that it does not respond to normal cells, while several mutations in cancer cells result in the expression of tumor-specific antigens that can be identified as non-self and activate the immune system, finally resulting in the elimination of cancer cells. The term “antitumor immunity” defines the innate and adaptive immune responses that regulate tumor. Both innate and adaptive immunity play a role in the identification and fight against tumors, and a successful antitumor immune response is related to the close interaction of several factors of innate and adaptive immune responses [13]. They are composed of antigen-presenting cells, various subsets of T cells, B cells, and NK cells. However, tumors use several ways of immunosuppression to stop the antitumor effect of immune cells. Dysregulation of the balance between the effector and regulatory cell compartments is one of the key strategies for tumors to escape immune eradication [14]. A better understanding of the vital immune cells and the regulatory networks participating in the interaction between tumor cells and the immune system is central for the improvement of therapeutic strategies to strengthen the immune system against cancers.

Numbers of cells die every day as a result of normal tissue turnover that is central for homeostasis maintenance in organisms [15]. Therefore, the existence of several forms of cell death is not unexpected [16,17]. Cell death can be categorized according to its morphological appearance, enzymological criteria, functional features, or immunological properties [18]. Classification based on morphological criteria proposes the existence of three various forms of cell death [19].

Type 1 cell death, or apoptosis, is described by some characteristic morphological changes such as condensation of nuclear material, DNA degradation, cell shrinkage, membrane blebbing, and presence of apoptotic debris [20]. Apoptotic cell death occurs constantly in multicellular organisms and is crucial for normal growth, tissue homeostasis, and many other physiological functions [21]. For supporting the host, physiological apoptosis is quickly recognized by phagocytic cells such as macrophages and dendritic cells (DCs) [22]. Phagocytic clearance of apoptotic bodies is a silent process [3]. These apoptotic cells get removed without inducing immunological responses due to the release of anti-inflammatory signals [23]. So, apoptosis has been introduced as an immunologically quiet form of cell death [3].

Type 2 cell death, or autophagy, is an intracellular degradation process that plays a central role in the protection of cells and organisms from stressors. Autophagy is described by the removal of materials marked for destruction into double-membrane vesicles called autophagosomes. Autophagosomes fuse with endosomes and then they deliver their content to the hydrolytic interior of lysosomes for degradation. Autophagic degradation is considered as a vital source of amino acids, nucleotides, and fatty acids and provides energy for cells. Autophagy plays a part in physiological processes and its activity is fundamental for adjustment to starvation, cell development, aberrant structures degradation, cell survival, homeostasis, and regulation of cell death [24].

Type 3 cell death, or necrosis, is generally characterized as a form of cell death with no indications of apoptosis or of autophagy. The morphological features of necrosis are cytoplasmic swelling, the plasma membrane breakdown, and loss of intracellular contents [25]. Dissimilar to apoptosis, the biochemical characteristics of necrosis are unknown [26]. Because of the abrupt release of pro-inflammatory modulators, necrosis is commonly believed to be immunologically unsafe [27,28]. Necrotic cell death leads to the release of pro-inflammatory mediators [29] (Figure 1).

Poorly immunogenic cancer cells avoid destruction by the immune response. Cancer cells express antigens but these prevent separating them from tolerized self-antigens. Normally mutation rates are low in such cancers and exhibit poor de novo antigens [30]. Glioblastoma [31], ovarian cancer [32], and some other types of malignancies lack stimulatory tumor neoantigens and endorse an immunosuppressive tumor milieu through generating anti-inflammatory cytokines [6,33]. Some cancer therapies induce immunologically silent apoptotic cell death and can attenuate the immune system, facilitating cancer recurrence [6]. Unlike apoptosis, a regulated cell death process in which the plasma membrane does not rupture, necrosis is inevitably associated with the release of DAMPs and intracellular organelles. Pathophysiologically, this stimulates an immune response because new surfaces become available to both the innate and the adaptive immune system during a process defined as necroinflammation [34]. Instead, apoptosis was considered to be non-immunogenic or even tolerogenic. Currently, a set of anticancer therapies has developed that causes a kind of apoptosis named ICD, informing the immune system to the existence of dying cancer cells [35]. ICD describes a functionally distinct type of cell death that results in a T cell-mediated immune response, particularly for dead cell-derived antigens. ICD can be stimulated by different mechanisms and leads to translocation or release of DAMPs from the dying cells that activate the immune response. In contrast to ICD, tolerogenic cell death (TCD) including most types of apoptosis is a non-inflammatory mechanism of cell death that is described by membrane blebbing and loss of DAMP secretion. TCD is unable to stimulate an immune response associated with cell death [36].

It was documented that if specific treatments are used, the immunogenicity of cancer cells increases enough to stimulate long-term antitumor immunity [37]. The ICD induction can change dying cancer cells into “vaccines” to induce anticancer immunity by the DCs’ maturation and cytotoxic T lymphocytes’ (CTLs’) activation [38] plus increasing the cytotoxic function of NK cells [35]. Immunogenic features of ICD are mediated largely by intracellular molecules termed DAMPs, which are invisible in live cells, but they exhibit immunostimulatory effects after exposure to dying cells [11].

## 3. Major Hallmarks of ICD

ICD is a form of apoptosis characterized by the emission of particular molecules that are a class of the danger-associated molecular patterns (DAMPs) family. DAMPs are intracellular factors typically unseen in live cells, which obtain immunostimulatory features upon exposure or secretion by dying cells. This process can be stimulated by some anticancer therapies [4,39,40]. DAMPs may be divided into three main subclasses according to their stage and localization/release place: (1) DAMPs appear on cell surface (e.g., CRT, HSP 70, HSP 90), (2) DAMPs appear extracellularly (e.g., HMGB1, uric acid, and pro-inflammatory cytokines), and (3) DAMPs appear as end-stage degradation factors (e.g., ATP, DNA, and RNA) [3,19,41]. DAMPs are sensed by pattern-recognition receptors including toll-like receptors (TLRs), the NOD-like receptors (NLRs), and retinoic acid-inducible gene-I-like receptors, (RIG-I-like receptors (RLRs)) [42].

The key DAMPs include CRT, HMGB1, ATP, HSP 70 and 90, and type 1 interferon and ANXA1 [43,44]. These molecules, by sensing their receptors, recruit and trigger immune cells. Lastly, they lead to phagocytosis and activation of T lymphocytes to eradicate tumor cells (Figure 2) [41]. The capacity of DAMPs produced during ICD is dependent on endoplasmic reticulum (ER) stress, which co-occurred or was activated by reactive oxygen species (ROS) [8,45].

### 3.1. Calreticulin (CRT)

CRT, perhaps the best-identified DAMP, is a 46 kDa Ca^2+^-binding protein, which is normally in the lumen of the ER, has different roles like chaperone activity and maintenance of Ca^2+^ homeostasis [46]. CRT has three domains with different affinities for calcium-binding followed by an ER retention sequence [47]. CRT has several functions, which include regulating protein synthesis, cell proliferation, adhesion, invasion, and nuclear transport [48]. CRT participates in the correct collection of major histocompatibility complex (MHC) class I and in antigen presentation, enabling the phagocytosis of apoptotic cancer cells [5,49,50]. Gardai et al. revealed the role of CRT in the clearance of apoptotic cells [51] and introduced it as a recognition ligand on the surface of the apoptotic cells. Different studies indicate that CRT is a significant factor in the immunogenicity of dying cells and antitumor immunity [52]. It is confirmed that ER stress is a significant factor in the ICD process. ICD inducers generate a form of ER stress that can be sensed by the immune cells through the release or surface exposure of DAMPs including CRT. The cellular response to ER stress leads to the stimulation of a signaling network called unfolded protein response (UPR). The stimulation of the UPR is correlated with three ER stress sensors: Inositol-requiring enzyme 1 (IRE1), protein kinase R-like ER kinase (PERK), and activating transcription factor 6 (ATF6) [53]. In normal conditions, these proteins are maintained in an inactive state by the master regulator of the UPR, Glucose-regulated protein 78 (GRP78)/binding protein (BiP), while, when intra/extracellular stressors generate unfolded proteins in the ER, BiP separates from these ER stress sensors, thereby stimulating the UPR-signaling network. During ICD, while other sensors of the UPR are activated, only the PERK is so far compulsive for ICD. PERK dimerizes, autophosphorylates, and in turn phosphorylates eukaryotic initiation factor 2A (eIF2a) that stops protein synthesis to reduce the ER protein synthesis amount [54]. CTR exposure in response to ICD inducers needs downstream ER stress, caspase-8-based cleavage of the ER-resident protein, B cell receptor–associated protein 31 (BAP31), and stimulation of BCL2-associated X protein (Bax) and Bcl-2 homologous antagonist/killer (Bak). ER-derived CRT is exocytosed through a classical, soluble N-ethylmaleimide-sensitive fusion protein-attachment protein receptor (SNARE)-dependent pathway. Exocytic vesicles originating from the Golgi have to fuse with the plasma membrane by molecular interactions between vesicle-related SNAREs and plasma membrane-related SNAREs (Synaptosome Associated Protein (SNAP)23/25) [55]. Along all the way from the ER to the plasma membrane, CRT is accompanied by ERp57. This interaction is necessary for CRT translocation because the deletion of ERp57 stops CRT exposure [56]. It is noteworthy to state that upon treatment with Type II ICD inducers fewer demands are required, since this pathway only depends on PERK, Bax, Bak, and the secretory pathway [53]. ER stress results in the exposure of CRT on the outer membrane of the cell surface [4], and CRT binds to receptors (such as cluster of differentiation 91(CD91)) on the DC surface, enabling the phagocytosis of dying cancer cells [57]. Ecto-CRT functions mainly as an “eat me signal” and activates antigen presenting cells (APCs) [52,58]. Obeid et al. confirmed that ecto-CRT is involved in anthracycline- and mitoxantrone-induced apoptosis and revealed the positive association between CRT exposure and the inducers-mediated immunogenicity [52]. Blockade of CRT by Small interfering RNA (siRNA) or antibody in colon cancer cells compromised the immunogenicity of mitoxantrone-treated CT26 mouse, while the administration of recombinant CRT restored it to the cell surface. [59]. Additionally, a study revealed the effect of CRT mRNA expression in tumor cells on the density and composition of immune infiltrative cells. Particularly, CRT expression associates with DC and CTL infiltration in different cancers including breast and ovarian cancers [60]. Recently, it was revealed that miR-27a targets CRT in colorectal cancer. The results described miR-27a as a negative regulator of the drug-induced ICD response by decreasing CRT expression levels and cell surface exposure [61,62]. These data indicated that CRT exposure on tumor cells in the ICD event facilitates their recognition by DCs and CTL responses [55,63,64] (Figure 3).

### 3.2. Adenosine Triphosphate (ATP)

ATP is another DAMP that is released from cells in the ICD event [64,65]. ATP participates in several cellular metabolic functions and intracellular response. It has detected that secretion of the ATP from dying cells is crucial for effective initiation of ICD [7,9,66]. ICD inducers lead to ATP release through a lysosomal-associated membrane protein (LAMP) 1 dependent mechanism, that translocates to the cell surface in a caspase- and pannexin 1 (PANX1)-mediated process. PANX1 is necessary for ICD-related translocation of LAMP1 to the cell surface and also for lysosomal exocytosis [67]. Extracellular ATP facilitates strong chemotactic effects by binding to purinergic receptor P2Y2 (P2RY2) and purinergic receptor P2X7 (P2RX7), respectively, on APCs and their precursors [68,69]. So, the dying cell’s immunogenicity is canceled not only when ATP fails to gather in the microenvironment of these cells [68,70], but also when P2RY2 or P2RX7 are absent from the myeloid part of the immune system. P2RX7 signaling activates the NLR domain-containing protein 3 (NLRP3) inflammasome, which in turn induces the secretion of active IL-1*β*, a central cytokine for the extension of antitumor immunity [69]. Studies showed that secreted ATP was able to intensify the motility of DCs [71] and tumor cells [72]. Furthermore, the maturation of DCs and the expression of the other molecules like CD40, CD80, and CD86 can be supported by ATP [73]. The apoptotic phase of cell death that stimulates ICD can be important in the secretion of ATP [8].

### 3.3. High Mobility Group Box 1 Release (HMGB1)

HMGB1, one of the DAMPs, was introduced in 1973 and is a nonhistone chromatin-binding protein [74,75]. HMGB1 is the first identified member of the HMGB family [76]. The HMGB family are three proteins, HMGB1, HMGB2, and HMGB3, that have a common structure [77]. They are structured into three domains, two homologous DNA-binding domains, boxes A and B, and an acidic C-terminal ending [78]. Subcellular localization of HMGB1 determines its function [2]. In the nucleus, it plays various roles including maintenance of nucleosomes, DNA repair, recombination, and transcription [79], while the extracellular HMGB1 mediates several biological functions including angiogenesis and wound healing as well as chemotaxis. [80]. In the extracellular environment, HMGB1 signals danger to the adjacent cells, activates inflammation, and induces immunity by binding to various receptors [81,82]. HMGB1 can be released from immune cells by active secretion or passively released from injured or dead cells, and leads to inflammation by attaching to several receptors [59]. Receptors for HMGB1 are RAGE (receptor for advanced glycation end-products), TLR2 and TLR4, and other receptors [59]. Once released during death, HMGB1 functions as a DAMP, regulating the inflammatory and immune responses. It has been revealed that some anticancer factors and stimulators of apoptosis and ICD can support HMGB1 release [83]. Binding of extracellular HMGB1 to the TLR4 on immature DCs stimulates a myeloid differentiation primary response (MYD) 88-dependent signaling pathway that leads to DC maturation cytotoxic T cells’ activation [84,85].

### 3.4. Heat-Shock Proteins (HSPs)

Another ICD marker, heat-shock proteins/chaperones, are responsible for protein folding and gathering and can support protein refolding under stress conditions [86]. Depending on their intra/extracellular location, HSPs either apply protection against environmental stress or function as powerful inducers of the immune response. As danger signals, intracellular HSPs overexpress and translocate to the plasma membrane under stress conditions like oxidative stress, irradiation, or during exposure to chemotherapy agents [87]. HSPs are categorized into two groups [88,89], high molecular weight (HMW) and small molecular weight (SMW) HSPs [86]. HMW-HSPs need co-chaperones for their action, so they are ATP-dependent chaperones including HSP90, HSP70, and HSP60 [74]. While SMW-HSPs are ATP-independent chaperones like HSP27 [90]. Between numerous HSPs, HSP70 and HSP90 are mainly related to ICD [91]. HSP70 and HSP90 show a dual function in cancer. Intracellular HSPs have a defensive function. Their anti-apoptotic characteristics can be the reason for the cytodefensive functions of HSPs [87]. In opposition to intracellular HSPs, extracellularly located or membrane-bound HSPs show immunological properties [92]. By moving the intracellular HSPs to the plasma membrane surface, they show powerful immunostimulatory function [93]. For example, HSP70 and HSP90 show immunostimulatory function when exposed on the extracellular layer of dying cells [3]. Ecto-HSP70 and HSP90 function as DAMPs and control the immunogenicity of dying cells [94] because they can interact with some receptors on APCs’ surface [95], like CD91, LOX1, and CD40 [93], and enable cross-presentation of tumor cells’ antigens to MHC class I molecule and activate CD8+ T-cells [96,97]. Altogether, it was revealed that the surface appearance of HSPs on tumor cells facilitates starting effective anticancer immune function by promoting the immunogenicity of tumor cells [2].

### 3.5. Type I IFN

Type I IFN plays a significant role in the ICD process. Besides the important character of type I IFN in the innate and adaptive immune responses [98,99], recently it has been recognized as a necessity for ICD in tumor cells [59]. Type I IFN not only stimulates antigen presentation but also has a role in the promotion of antitumor T cell [100,101]. ICD-stimulating factors, including anthracyclines, radiation therapy, and oncolytic viruses, have been revealed to increase the expression of type I IFN response genes in the tumor cell [100,102,103]. Among the most bioactive IFN-I-induced genes, the chemokine (C-X-C motif) ligand 10 (CXCL10) functions as an important chemotactic factor and recruits immune members that selectively attack the tumor [101].

### 3.6. Annexin A1 (ANXA1)

ANXA1 is a member of the calcium- and phospholipid-binding protein family. It was revealed that myeloid cells, lymphoid cells, and several epithelial cell types express ANXA1 [104]. It was described that ANXA1 is helpful to DC function during ICD due to its immunosuppressive activities (81). To facilitate the interaction of APC and dying cancer cell, ANXA1 released from the apoptotic cells attaches to formyl peptide receptor 1 receptor on APCs [105,106]. So, ANXA1 mediates tumor antigen uptake for presentation [107]. It was confirmed that, in cancer cells without Annexin-A1 (ANXA1) and immune cells without formyl peptide receptor 1 (FPR1), anthracycline or oxaliplatin failed to show therapeutic effects [108]. In head and neck squamous carcinoma, overexpression of the miRNA-196 leads to reduced expression of ANXA1 [105]. This miRNA expression is regulated by DNA methylation in several cancers [109]. Suppression of miRNA-196 expression leads to increased ANXA1, permitting appropriate ANXA1 production during ICD stimulation [107].

## 4. ICD Inducers

Only certain cancer treatments have been reported to stimulate ICD. It has been suggested that these immunotherapeutic modalities can be divided into two groups, type I and type II ICD inducers. The majority of ICD inducers are considered as type I inducers of ICD but only a scant number of type II ICD inducers have been recognized [45]. The two systems of ICD inducers exploit different pathways of cell death induction according to their ability to activate ER stress response and subsequent production of ROS in a direct or indirect manner. Treatments that belong to the first category are known to target subcellular compartments like cytosolic proteins, plasma membrane proteins/channels, DNA or proteins of DNA repair, and/or replication machinery while the ER is affected as a ‘collateral’ effect [45]. On the contrary, type II ICD inducers are considered to lead cancer cell death by direct acting on ER [8,110]. Being a cell factory for protein production and assembly, the ER network possesses a precise and rigorous quality control system called (ERQC) for final monitoring of the synthesized biomolecules. When subjected to biotic and abiotic environmental stressors, the levels of aberrant unfolded/misfolded proteins may overwhelm the quality control systems thus allowing protein aggregates to be formed in the ER. The resulting imbalance between the cellular demand for and capacity of ERQC is referred to as ER stress. When ER stress is prolonged or intense it can cause ER and intracellular organelle damage or cell death in extreme cases. Accumulation of aberrant proteins triggers the unfolded protein response (UPR), which in turn potentiates ERQC activities to reduce the burden of misfolded and unfolded proteins entering the ER [111].

The ER is involved in a network of signaling events that respond to unfolded proteins loaded in its lumen. The unfolding of proteins provides a molecular threat to living cells [112], which triggers a system of signaling pathways in cells, the condition that is called the unfolded protein response (UPR) [113]. In addition, a number of insults that interrupt protein folding within the ER stimulate the UPR [114]. Persistent activation of the UPR pathway eventually can cause programmed cell death mainly through apoptosis [115]. ER stress-induced apoptosis is a fundamental response, which ensures a cellular protective mechanism. Chemotherapeutic drugs such as anthracyclines, oxaliplatin, and cardiac glycosides indirectly induce ER stress, whereas physical modalities belong to type II inducers primarily initiate ER stress and ultimately lead to cancer cell death. Physical cell death-inducing modalities, such as photodynamic therapy (PDT), have been confirmed to be able to act as in situ vaccines and to aid in inducing antitumor immunity in human patients (PDT), most probably by inducing ICD. Of note, cancer treatment might take more advantage of ICD induced by physical modalities rather than chemotherapeutic-based ICD, due to the fact that chemical drugs have been detected to leave behind active drug debris [116]. The two classes of ICD inducers are described in more detail below (Table 1).

### 4.1. Type I ICD Inducers

#### 4.1.1. Anthracyclines

Anthracyclines are a class of widely used antitumor treatments used for lymphoma, leukemia, ovarian, uterine, and breast cancers [125]. They are believed to work by intercalating into DNA and suppressing DNA topoisomerase II activity [126] and thereby trigger cell death. The apoptotic cell death stimulated by anthracyclines has been described with potent immunogenicity that is related to the emission of early and late danger signals including CRT exposure, ATP secretion, and HMGB1 release [45]. Doxorubicin, as a member of the anthracycline family, was the first immunogenic inducer identified, in 2005 by Kroemer and Zitvogel [127]. Induction of DAMPs on tumor cells undergoing death is required to incite the recruitment of DCs for eliciting further immunity by T cells toward dying malignant cells [52]. Of note, it has been reported that immunogenic death induced by doxorubicin (tested at five concentrations (0.1–10 μmol/L)) does not directly stimulate the activation of APCs [128]. Administration of doxorubicin by intraperitoneal injection in mice showed that noncancerous cells undergo ICD in response to doxorubicin (injected at 10 mg/kg) that was associated with TLR-2/TLR-9-MyD88-dependent signaling, leading to the inflammatory response [129]. It should be investigated whether this molecular sensor functions in antitumor response or not.

Neoplastic cells exposed to anthracycline-driven ICD, present ecto-CRT at pre-apoptotic phase, once any signs of morphological apoptosis do appear [130]. The secretion of ATP, which is a key event of ICD, occurs concurrently with the exposure of phosphatidylserine on dying cancer cell lines responding to anthracyclines [131]. The active release of ATP, in contrast to CRT exposure, has been found in a tight connection with autophagy induced by anthracyclines since, in mice deficient in autophagy, no ATP was released from dying tumor cells [65]. Analysis of tumor tissues after anthracycline therapy in mice demonstrated the infiltration of IL-17-producing γδ T lymphocytes (γδ T17 cells) and subsequent invasion of CTLs into the tumor microenvironment (TME). It was shown that suppression of γδ T17 cells by the knockout of T cell receptor δ or Vγ4/6 inhibited the production of IL-17 by T cells surrounding the tumor bed and abrogated the therapeutic efficacy of anthracyclines. Accordingly, IL-17A–IL-17R pathway inhibition led to a reduced tumor-specific T cell response, underscoring a significant role for γδ T17 cells in anthracycline-induced immunity [132].

#### 4.1.2. Cardiac Glycosides 

Cardiac glycosides are a group of natural compounds used in treating heart failure that influences Na^+^/K^+^ balance by targeting Na^+^/K^+^ pump in cells [133]. The beneficial effects of cardiac glycosides (at a dose of 0.5 mg/kg) have also been associated with antitumor immunity [117]. Evidence has shown that they are able to induce the ER stress pathways involved in the SRC kinase-EGFR- Mitogen-activated protein kinase (MAPK) pathway and further mitochondrial ROS production [117,134]. Moreover, upregulation of death receptor (DR)-4 and -5, T-lymphoma invasion and metastasis (TIAM1), and Protease-activated receptor 4 (PAR4), as well as inhibition of DNA topoisomerase that are all mediated by the pro-apoptotic effects of cardiac glycosides, provide additional mechanisms which contribute in growth arrest of tumor cells [134]. Of interest, two compounds of cardiac glycosides, digoxin, and digitoxin have been shown to induce ICD in cancer cells. They effectively induce characteristic signatures of ICD such as CRT exposure, ATP secretion, and passive release of HMGB1 [117]. In vivo experiments revealed their effectiveness at tumor regression. To establish that the immune system was ignited by cardiac glycosides, murine fibrosarcoma cells were treated with digoxin and digitoxin in combination with cisplatin, mitomycin C, or mitoxantrone and the immunogenicity of antitumor drug-treated cancer cells was evaluated in vivo in mice. A good percentage of tumor-free mice were attained after rechallenge with live tumor cells in mice pre-injected with tumor cells treated with either digoxin or digitoxin. These results showed that both digoxin and digitoxin can promote antitumor immune response comparable to mitoxantrone, which is a potent ICD inducer [117].

#### 4.1.3. Capsaicin (CPS)

Another cytotoxic anticancer therapeutic is CPS, a neurotoxin homovanillic acid derivative. CPS as a major constituent of peppers has been shown to interact with a nonselective cation channel called the transient receptor potential vanilloid type-1 (TRPV1) [135]. Short-term exposure to capsaicin can induce cell death in normal and various types of malignant cells through TRPV1-mediated signaling [136,137]. In the study of the effects of CPS in antitumor immunity, human bladder cancer cell lines undergoing apoptosis (dose-dependent apoptosis reaching approximately 30% apoptotic cells using 150 μM CPS) were shown to emit ICD hallmarks after treatment with CPS in vitro [118]. CPS exerts apoptosis in various human cancer cells including respiratory, urogenital, skin, and gastrointestinal malignancies [138,139,140,141]. The molecular events characteristic of CPS-induced ICD include ectopic expression of CRT and HSP90, the early active release of ATP, and the extracellular passive release of HMGB1, HSP70, and HSP90 [118]. Although the mechanisms underlying CPS-associated ICD are not clearly known, it is thought to promote ROS generation and ER-dependent stress [141,142,143]. Furthermore, the CPS-induced mitochondria-mediated cell death pathway has been reported to be involved in this effect.

#### 4.1.4. Clostridium Difficile Toxin

Bacterial toxins have already been discovered to induce immunogenic death of cancer cells [144,145]. Clostridium difficile is a disease-causing bacterium with potent pro-apoptotic and pro-inflammatory activities capable of developing antitumor immunity [146,147]. The virulence factor of Clostridium difficile toxin B (TcdB) was found to trigger immune responses against cancer cells in vitro and in vivo [148]. TcdB has been shown to induce cell death through apoptosis or necrosis in target cells [149,150,151]. Inside the target cells, TcdB acts on Rho proteins in order to modify Rho protein activity, which appears to constitute the molecular basis of TcdB-induced cell death [152,153]. It was shown that toxin-treated (500 ng/mL of TcdB) mouse colorectal cancer CT26 cells were immunogenic and caused mouse bone marrow-derived dendritic cell activation in vitro. Mice immunization with intoxicated CT26 cells elicited cell-mediated tumor-specific immunity that supported mice against the same tumor cells challenge [148]. Further experiments identified the mechanisms involved in ICD induction by TcdB. It was demonstrated that TcdB-induced cell death involved an increase in the intracellular ROS that was confirmed by the use of chemicals that act to attenuate ROS levels [119]. In addition, CRT exposure, secretion of ATP, HSP70, and HSP90, and HMGB1 release and autophagy were among the hallmarks of apoptosis induced by TcdB in intoxicated tumor cells [119]. The antitumor effects of TcdB implicated it as a good candidate in generating chemotherapeutics with immunogenic properties to fight against cancer. However, for successful employment of TcdB in clinical settings, one could overcome the obstacles, such as hard assembly to immunotoxin and specific targeting of cancer cells, inhibiting side effects that are big challenges.

#### 4.1.5. High Hydrostatic Pressure (HHP)

Further experiments have described HHP as a new treatment modality capable of stimulating immunogenic apoptosis. In the field of biotechnology, HHP is used in sterilization procedures of foods and human transplants [154]. HHP was described as a treatment modality for cancer patients in 1972 [155]. The first indications for its implication in ICD induction were achieved from studies in mouse models [156,157]. HHP affects human cell viability [158] and has been suggested as a physical technique to manufacture therapeutically efficient cell-based tumor vaccines [158]. Fucikova et al. extensively assessed the effects of HHP treatment in both primary tumor cells and tumor cell lines of human prostate and ovarian cancer and in acute lymphocytic leukemia cells [159]. Similarly, to anthracyclines, HHP rapidly upregulates the expression of ICD markers (CRT, HSP70, and HSP90) on the cell surface and activates ATP release into the extracellular environment. The killing of tumor cells by HHP provides a stimulatory impact to mature monocyte-derived DCs, as was measured by the upregulated expression of maturation markers on DCs and induced phagocytosis of tumor cells [120]. Additionally, the interaction of HHP-killed tumor cells with DCs influenced cytokine production in DC. The pathways engaged in the synthesis of IL-6, IL-1, IL-12p70, and TNF-α in DCs were affected by HHP-treated tumor cells. The use of immunogenic HHP-treated tumor cells to generate a DC-based anticancer vaccine in order to induce cancer-specific immune reactions has been initiated in clinical trials for ovarian and prostate cancers [160]. Treatment with HPP-killed tumor cells seems to be easily performed to produce a standardized protocol for cancer immunotherapy [116].

#### 4.1.6. Ultraviolet Light (UV)

Moreover, it has been shown that immunocompetent mice vaccinated with ultraviolet light (UV) C-irradiated tumor cells were immune to subsequent challenge with live cells, whereas immunodeficient mice developed cancer [161]. UV is electromagnetic radiation longer than X- and γ-rays. UVC refers to short wavelength UV radiation (200–280 nm), different from the range of 400–320 nm UVA and UVB (320–280 nm). UV light irradiation of cells can cause DNA damage and the affected cell consequently undergoes apoptosis or necrosis [162]. However, only UVC-mediated cell death, but not UVA and UVB radiation, was shown to possess immunogenicity, which is associated to the translocation of CRT to the cell surface at early apoptosis and HSP70 and HMGB1 release later on [59,121]. All the UVA, -B, and -C lights have the ability to produce ROS and contribute to cellular stress response pathways [163,164], hence the enigma of the inability of UVA and UVB to induce ICD, much remains to be explored. The inflammatory response mediated by CD8+ T cells and NK cells has been shown to participate in protective immunity caused by UVC-treated cancer cells. DCs loaded with UVC-treated cells displayed a strong phagocytosis activity and maturation, which in turn drives cytotoxic IFN-γ, producing CD8+ T cells responses [121]. Second, DCs challenged with UVC-died tumor cells were characterized with increased expression of genes involved in antigen processing and pro-inflammatory cytokine production [45]. To date, there is no clinical evidence of UVC treatment in human tumors, since UVC light can cause a high rate of putative pro-oncogenic mutations leading to tumor progression [116].

### 4.2. Type II ICD Inducers

#### 4.2.1. Photodynamic Therapy (PDT)

Phototherapy or the use of light to cure disease has been established in ancient Indian, Egyptian, and Chinese civilizations [165]. The traditional phototherapy turned to modern photodynamic therapy (PDT) as a result of experiments on the combination of light and chemicals for better therapeutic efficiency [166]. Using a chemical component that acts as a photosensitizer indicates that PDT is considered a physicochemical, rather than an exclusively physical, antitumor modality. Exposure of cells to PDT starts a chain reaction leading to the production of ROS and ultimately cell death caused by oxidative stress. Studies during the late 1980s and early 1990s also demonstrated the recruitment of inflammatory cells such as macrophages, leukocytes, and lymphocytes into the tissues treated with PDT [167,168], indicating activation of the immune system. Upregulation of pro-inflammatory cytokines, IL-6 and IL-1 but not TNF-α, have also been reported as a result of photodynamic therapy [169]. Comparing tumor growth in immunodeficient and immunocompetent Balb/C mice showed that long-term effects of PDT therapy were attributed to the activation of the antitumor immune response as tumor development frequently occurred in deficient but not in normal mice, whereas short-term responses were similar in both normal and immunocompromised mice [170]. The anticancer properties of PDT are relevant to direct impacts on cancer cell death and its immunogenic features. PDT exerts cytotoxic activity against malignant cells through activating antitumor immunity in general and by inducing bona fide ICD [12]. Unlike chemotherapy, very little is known about the molecular mechanisms that account for the immunostimulatory functions of PDT. It has been identified that the immunogenic demise driven by PDT exposure is accompanied by CRT, HSP70, and HSP90 translocation from ER to the plasma membrane and extracellular secretion of HMGB1 and ATP [171,172]. PDT based on the photosensitizer hypericin preferentially targets the ER and induces ROS formation adjacent to the ER [173]. Hypericin predominantly causes ER-associated oxidative stress and initiates different signaling pathways of UPR, which finally leads to amplified mitochondria-dependent apoptosis [174]. Kinetically, the emission of DAMPs (ecto-CRT, HSP70, and secreted ATP) in response to hypericin-based PDT did occur more quickly than those previously reported for these DAMPs [171]. Importantly, hypericin (Hyp)-PDT has several advantages over classical anticancer regimens [45]. ICD induced by Hyp-PDT was much more effective when compared to that induced by chemotherapy or radiotherapy. Specifically, using Hyp-PDT prevents tumorigenesis by impeding tumor-promoting cytokines signaling and downregulating mediators of tumor metastasis-like cancer-derived matrix metalloproteinase-9 (MMP-9). It has been reported that Hyp-PDT strongly inhibits transcriptional activities of nuclear factor kappa-light-chain-enhancer of activated B cells (NF-kB) and Activator protein 1 (AP-1) in treated cancer cells. Moreover, Hyp-PDT can reduce the cancer cell-secreted tumor-promoting cytokines such as Granulocyte-macrophage colony-stimulating factor (GM-CSF), IL-6, and TNF [175]. Although there have been many clinical trials of PDT treatment in various cancers [12], more research is needed to determine all the functional features of Hyp-PDT.

#### 4.2.2. Pt-NHC

It has been known for many decades that N-heterocyclic carbene (NHC) complexes are potent antiproliferative compounds used in medicine [176]. Recent work has shed new light on the ability of a unique cyclometalated complex bearing platinum Pt-NHC to elicit ER-associated production of ROS. Wong et al. screened a library of Pt agents including oxaliplatin, carboplatin, cisplatin, and their Pt^IV^ prodrugs for their ICD-inducing features [122]. Biochemical hallmarks of ICD by Pt-NHC were clearly demonstrated to include surface-exposed CRT and the extracellular release of ATP and HMGB1 [122]. It has been demonstrated that Pt-NHC selectively localizes in the ER and enhances ER stress-mediated cell death indicating that Pt-NHC is a type II ICD inducer [177].

#### 4.2.3. Coxsackievirus B3

Moreover, within the broad spectrum of oncolytic viruses, coxsackievirus B3 (CVB3) has been well-established to generate an immunogenic TME (e.g., by increasing the CD8+ lymphocytes and development of tumor-nourishing blood vessels) [123]. More analysis proved that treatment with CVB3 induces a molecular profile indicative of ICD induction [124]. Virus replication into the host cells can cause considerable accumulation of viral proteins in the ER, which leads to ER stress and finally cell death [45]. In addition, measles virus, Newcastle disease virus (NDV), and herpes simplex virus (HSV) are found to possess an immunostimulatory nature toward a variety of malignancies [178,179,180]. However, the mode of action of these viruses in the induction of ICD should be investigated.

## 5. Subversion of ICD

The molecular pathway of CRT exposure involves drastic disorganization of ER structure, which is accompanied by PERK phosphorylation as a sign of ER stress response. The blockade or knockdown of PERK has been shown to abolish the translocation of CRT to the cell surface [55], indicating that the function of PERK dictates CRT exposure. This inflammatory kinase accordingly phosphorylates downstream effector eIF2α on serine 51 that is indicative of its inactivation [55]. The eIF2 is a GTP-binding heterotrimeric protein responsible for the initiation of translation in eukaryotes [181]. This interaction leads to the arrest of protein synthesis in response to certain types of stress such as UPR and viral infections [182]. A spectrum of viral factors has been shown to prevent eIF2α phosphorylation [183]. This effect is mediated through a plethora of direct or indirect mechanisms of action. One significant mechanism they act upon is disrupting eIF2α binding and phosphorylation by the release of products acting as a pseudosubstrate/mimic inhibitor of PKR. For example, the TAT protein from human immunodeficiency virus 1 (HIV1), M156R from myxoma virus, C8L from swinepox virus, and hepatitis C virus envelope protein E2 [184,185,186,187] are contained in this group. The other operational strategies include direct inhibition of PKR⁄PERK (exploited by γB glycoprotein from herpes simplex virus type 1 (HSV1), non-structural protein 1 (NS1) protein from influenza virus, SM from Epstein-Barr virus (EBV), vIRF-2 from Kaposi’s sarcoma-associated human herpes virus 8 (KS-HHV8), and the vaccinia virus E3L product [188,189,190,191,192]), cytoplasmic relocalization of PKR to the nucleus or other subcellular compartments (by polypeptides binding to PKR from human papillomavirus (HPV), murine cytomegalovirus (mCMV), and hCMV [193,194,195]), restricting PKR activation by binding or inhibiting double-strand RNA production (e.g., E3L from vaccinia virus and m142 and m143 proteins from murine cytomegalovirus [188,196]), and generation of viral activators of PKR inhibitors like p58 [197,198].

The disulphide isomerase ERp57 constitutes an element of the pre-apoptotic pathway that participates in the appearance of CRT on the surface of dead cells that are treated with specific types of anticancer agents [55]. Both CRT and ERp57 are synthesized and collected in the lumen of the ER and are usually presented at the surface of cells subjected to ICD [52]. Cells that lack the ERp57 cannot precede chemotherapy-induced ICD because they are unable to expose CRT and, hence, fail to stimulate an immune response [199]. On the other side, the depletion of CRT could suppress ERp57 exposure [199]. These data demonstrate that ERp57 is requisite for the exposure of CRT and vice versa. Brefeldin A, (i.e., blocker of anterograde protein transport) contributes to the efficacy of chemotherapy by inhibition of pathways of ER-Golgi protein traffic [55]. The effects of brefeldin A on protein transport blocks translocation of CRT/ERp57 complex in murine CT26 colon cancer cells treated with anticancer agents, such as oxaliplatin, anthracyclines, or UVC light, and, accordingly, the potential of CT26 cells to immunize mice against tumor development in vivo after anthracycline treatment was destroyed in the presence of brefeldin A [55]. Moreover, the secretory route from the Golgi apparatus to the plasma membrane has been demonstrated to employ the molecular interactions between the synaptosomal-associated protein (SNAP) proteins and the respective plasma-membrane associated receptors SNAREs. Defect in SNAREs prevents the surface expression of CRT [200] and RNA interference targeting SNARE-based exocytosis resulted in reduced capacity of cancer cells incubated with anthracyclines, oxaliplatin, or UVC light to induce protective immunity in vivo. The depletion of SNAP23/25 did not have a profound impact on eIF2a phosphorylation, caspase-8 activation, and BAP31 cleavage, underscoring that the ER stress and apoptotic machinery work upstream of CRT/ERp57 co-translocation [201]. In this regard, the TAT protein of HIV1 has been shown to disrupt the intracellular transport of proteins via miRNA-mediated downregulation of SNAP25 in neurons [202].

The involvement of caspases in the immunogenicity of cancer cell death has been established as the treatment of cancer cells with general caspase inhibitors or p35 protein of baculovirus (known to block caspase activity) could abolish the location of CRT at the cell surface [127]. The caspase-dependent mechanism of ICD induction implicates caspase-8 but not the executioner caspase-3 and -7 or ER-resident caspase-12, which is demonstrated by the experiments that gene knockout models for caspase-8 prevented CRT exposure elicited by oxaliplatin, anthracyclines, or UVC light [55]. Further investigations into the correlation of PERK-mediated eIF2α phosphorylation and caspase-8 activation revealed that caspase-8 operates as a downstream component of the PERK pathway [55]. The proteolytic activation of caspase-8, in turn, leads to the cleavage of a Bcl-2-binding protein located at the ER, Bap31, whose function impairs protein transport from the ER to the Golgi complex and participates in response to ER stress [203,204]. Bap31 is required for the conversion of endoplasmic CRT to the plasma membrane CRT as the knockdown of Bap31 in HeLa cells or mutation of Bap31 to render it uncleavable similarly caused disruption of CRT exposure. One of the strategies applied by the immune system to defend against viral infections presumably lies in the programmed death of infected cells to restrict ongoing viral replication [205]. Evolutionarily, viruses have adopted strategies to overcome host cell death. Ideally, these adjustments particularly target multiple components in the apoptotic cascade initiated for ICD induction, for instance, caspase-8, Bap31, Bax, and Bak. Studies have further supported this conception by the findings that viruses have developed strong modules to block caspase-8 activation through direct caspase/serine protease inhibitors (of the serpin superfamily), which are exerted by SERP2 from myxoma virus and cytokine response modifier A (CrmA) from cowpox virus and orthopoxvirus [206,207,208,209] as well as by specific inhibitors of caspase-8 which directly interact with pro-caspase-8 including the viral inhibitor of caspase activation (v-ICA), produced by hCMV and E6 protein from HPV16 [210,211]. Like pathogens, malignant cells have been demonstrated to limit the immunogenicity of cell death induced by anticancer therapy. Advanced tumors escape immune recognition by suppression of the emission of DAMPs or detection of DAMPs by immune cells [212]. In one set of mechanisms, cancer cells appear to resist cell death by controlling the balance of pro- and anti-apoptotic members of Bcl-2 family in a way that is linked with overexpression of anti-apoptotic regulators or downregulation of their pro-apoptotic counterparts [213]. Additionally, the loss of TP53 is the most common process to circumvent apoptosis [213]. TP53 is the major part of a canonical signaling network involved in suppression of proliferation [214]. Focusing on the intercellular events in the elimination of dying tumor cells, it was found that the absence of HMGB1 expression can attenuate DC-dependent T cell responses toward tumor-associated antigens [215]. In line with these data, previous reports have shown that the efficacy of chemotherapy or radiotherapy is largely confined in patients with breast cancer who carry certain alleles of TLR4 causing loss of function mutations [59]. TLRs expressed by DCs have already been described to detect endogenous danger signals, like nuclear HMGB1 emanating from dying cells, and their function determines the success of anticancer therapeutic protocols [216]. Likewise, there are diverse strategies to subvert ICD such as polymorphisms within Interferon Alpha and Beta Receptor Subunit 1 (IFNAR1) [59], the weak expression of interferon regulatory factor IRF7 [217] and low Signal transducer and activator of transcription 1 (STAT1) levels [218] which require more detailed studies.

## 6. ICD and Combination Therapy in Cancer

Considering the fact that a significant number of therapeutic agents such as chemotherapy, γ-irradiation, and photodynamic therapy have widened the scope of immunogenic cell-death inducers due to having immune-modulatory or immune-stimulating effects, an urgent need for further investigations to determine their possible links with exposure or release of known/unknown DAMPs is greatly felt. It seems that ICD inducers will be ideal when used in combination with other therapies through involving various overlapping death pathways (Table 2).

### 6.1. Therapies Combining Chemotherapy-Induced ICD

There is a concurrence of opinions that chemotherapy can undermine the systemic immunity in patients via instigating an immunosuppressive state in both T cell and NK cell immunity, thus engendering infectious outcomes and germination of residual tumors [219]. On the other hand, numerous chemotherapeutic drugs or radiotherapy exert their cytotoxic impacts through the increased apoptosis response, and this apoptosis was commonly believed to be non-inflammatory and non-immunogenic. Moreover, antigen-specific T-cell immunity is commonly shown to be induced as a result of danger signals emitted by cells treated by chemotherapeutic drugs including oxaliplatin or radiotherapy [220,221]. If various chemotherapeutic agents are applied, a variety of DAMPs can mediate antitumor immunity; though the overall outcome is determined by the induction of immunogenic cancer cell death and the acquisition of tumor antigens by APCs once tumor-specific killer T cell response is elicited [93]. As opposed to the novel model of DAMP-defined immunogenicity of cell death, the immune-modulating effects of certain chemotherapeutic drugs were identified years ago and later triggered the derivation of the former concept. This is well substantiated by the fact that the immune-response modulatory role of doxorubicin long preceded its potential to induce surface exposure of DAMPs [2]. The adaptive immune response against dead cell-associated antigens is triggered by an immunogenic variant of apoptosis in immune-competent hosts when a number of chemotherapeutics are clinically employed [19]. Concerns over the safety issues of ICD inducers are greatly addressed, thanks to the approval of international regulatory agencies for the use of patients with hematological and solid neoplasms and also their incorporation in consolidated therapeutic protocols [222].

Suzuki et al. [220] revealed that after chemoradiation therapy and induced HMGB1 levels in esophageal squamous cell carcinoma (ESCC) patients’ serum, tumor antigen-specific T-cell responses in these patients were induced. What is more, according to the findings of immunohistochemical studies, preoperative chemo-radiotherapy and the degree of HMGB1 corresponded positively with HMGB1 production within the TME and patients’ survival, respectively. Moreover, HMGB1 high expression group upregulated the infiltrating grade of CD8(þ) T-cells within TME [220].

Cancer cell-specific activation mechanism of cyclophosphamide, a DNA alkylating agent belonging to the family of nitrogen mustards, had convinced scientists into believing that it can act as a cancer-selective drug; however, its cytotoxic effects on numerous cell types, including immune cells, soon got revealed [223]. Cyclophosphamide is utilized for wide-ranging clinical purposes, ranging from anticancer treatment of lymphoma, leukemia, and solid tumors to curing non-neoplastic immune diseases such as rheumatoid arthritis and multiple sclerosis [224]. Recently yielded data indicate that, in spite of the immunosuppressive functions of cyclophosphamide at high doses, impressive immune-stimulatory impacts can be attained from metronomic cyclophosphamide regimens, for instance, via either depleting or inhibiting Forkhead Box P3 (FOXP3) regulatory T cells (Tregs), and that is why cyclophosphamide are regarded as bona fide ICD inducers [44,225,226].

A class of anthracycline antibiotics consists of a natural compound called doxorubicin (also known as adriamycin) with the potential to function as a DNA intercalating agent. Once teaming up with other chemotherapeutics, like cyclophosphamide, it can combat certain types of leukemia, Hodgkin’s lymphoma, and myriad solid neoplasms, depicting relatively mild side effects [227]. The latest preclinical trials have affirmed the capacity of doxorubicin to trigger ICD as well as inducing anticancer immune responses [228].

It has long been recognized that doxorubicin increases the therapeutic potential of IL-12, TNF-α, and IL-2 in animal models of cancer [229]. A study conducted by Alagkiozidis et al. determined that combination therapy of doxorubicin with IL-18 at lower doses markedly induced the inhibition of tumor growth as compared with doxorubicin or IL-18 monotherapy. Increased expression of MHC class I and Fas by ID8 murine ovarian cancer cells and also sensitization to CTL and Fas-mediated killing were observed in vitro. In addition, they showed that combinatorial therapy made wide-ranging tumor suppression and survival rate in a large number of mice, whereas no survival was detected in mice treated with doxorubicin or IL-18 alone [230]. Another study was documented that mice with osteosarcoma tumors were treated with DCs and doxorubicin in combination and showed an increased frequency of CD8+ T cells in metastatic tumors and suppression of metastatic growth as well as increased expression of CRT and the release of HMGB1 from tumor tissues. Moreover, serum interferon-γ levels and systemic immune responses were enhanced in the treated mice [231].

Oxaliplatin, as a chemotherapeutic agent that developed in the 1990s, possessing so highly sophisticated molecular mechanisms that they could induce cytotoxic effects against cancer cells, generates DNA adducts and activates the pro-apoptotic transcription factor [232,233]. It was indicated that oxaliplatin but not cisplatin mediates its antitumor effects through an immunogenic mode of tumor cell death in colorectal cancer [234]. Immunogenicity of oxaliplatin was validated in a mouse model of vaccination, where CT26 colon cancer cells treated with oxaliplatin were used as a tumor vaccine to inoculate Balb/c mice. This treatment could stimulate an immune response and protect against the growth of CT26 tumors in immunocompetent mice [234]. Moreover, in conjunction with 5-fluorouracil and folinic acid, oxaliplatin can treat colorectal cancer in clinical practices. As well as inducing ICD, the expression of programmed death ligand-2 (PD-L2) is restrained by oxaliplatin and blocks immunosuppression effect by both DCs and tumor cells [234]. On the other hand, combinational therapies have been developed to fulfill ICD requirements of anticancer compounds lacking ICD-hallmarks. These strategies work to compensate for the missing characters of ICD induction [235]. Cisplatin drug has been the main area of focus in these studies. In this regard, the use of adjuvant agents in combination with cisplatin could generate full ICD features. Administration of zinc (Ⅱ) chloride, capable to induce ER stress, combined with cisplatin was able to enhance immunogenic effects in P53-deficient cancer cells [236]. After cessation of the chemotherapy process, residual tumor cells are able to upregulate adaptively the programmed death ligand-1 (PD-L1) so as to interact with the receptor of the immune checkpoint on the surface of T cells, which is a kind of escape system from immunosurveillance (immunosuppressive state). Therefore, there is a significant challenge in immunotherapy applications in cancers with immunosuppressive state [237]. It has been verified that monoclonal antibodies application for blocking of immune checkpoints could potentially provide high clinical achievement for cancer immunotherapy; however, their application is limited due to the cold environment of tumors [238]. Recently, many attentions have been focused to develop new and effective inhibitors for immune checkpoints in the immuno-oncology research area. Tumor cells could potentially escape from immunosurveillance and proceed with their specific mechanisms such as the pathway activation of immune checkpoints to repress the antitumor responses of the immune system. The inhibitors of immune checkpoints are able to revitalize the antitumor responses of the immune system through interfering with inhibitory effects on signaling pathways and also enhance the elimination of tumor cells by immune-mediated actions [239]. The implementation of such inhibitors against the PD-1/PD-L1 axis and probably CTLA-4 (cytotoxic T-lymphocyte antigen-4) has indicated considerable clinical benefits and even transformed rapidly as a practical method in medical oncology. Application of monoclonal antibodies, as potential co-inhibitors for immune checkpoints, particularly CTLA-4 and PD-1, has been utilized clinically for various malignancies, including bladder cancer, head and neck squamous cell carcinoma, Hodgkin lymphoma, non-small cell lung cancer, Merkel cell carcinoma, melanoma, microsatellite instability (MSI)-high colorectal carcinoma, and renal cell carcinoma [240]. These inhibitors also have improved the possible practical methods for cancer treatments in the oncology field. Ipilimumab, as the first confirmed inhibitor for immune checkpoints that targets CTLA-4 molecules, has been applied for the treatment of those patients who suffer from advanced melanoma. This inhibitor is able to prevent the inhibition of T cells and also promotes the proliferation and activation of T cells’ effectors. The other inhibitors, nivolumab and pembrolizumab that target PD-1, indicated promising results in the treatment of NSCLC (non-small cell lung carcinoma) and melanoma with the rate of objective response between 40 and 45%. Furthermore, the treatment of urothelial bladder cancer with inhibitors of PD-1/PD-L1 pathways revealed an increment in response rate in the range of 13 and 24% [239].

Different clinical experiments have recently demonstrated that the use of anthracyclines as an ICD inducer led to an increase in the blocking of PD-1 and PD-L1 efficacy. These data could support the beneficial concept of ICD in cancer treatment [241]. LTX-401 is a known conjugated-antibody T-PNU drug. Crizotinib, as an inhibitor for tyrosine kinase, and oxaliplatin, as a chemotherapeutic agent, have a natural potential to induce ICD. It was reported that the combination of LTX-401 and the inhibitors of CTLA-4 and PD-1 indicated considerable antineoplastic effects on primary distant and lesions tumors, showing the higher therapeutic effect of ICD and checkpoint blocker together [241]. A study showed that lurbinectedin treatment could induce ICD traits, such as calreticulin exposure, INF-I immune responses, ATP release, and the exodus of HMGB1 molecules, at in vitro condition. The anticancer and antineoplastic properties of lurbinectedin could be strengthened remarkably by its combination with the double blockade of CTLA-4 and PD-1 immune checkpoints [242]. In another study, conducted by Zhao et al., it was revealed that irreversible electroporation (IRE) combined with anti-PD-1 could significantly induce the ICD process and also mediate the consequence durable response in a treated model (orthotopic pancreatic ductal adenocarcinoma (PDAC)). The mechanistic investigations disclosed that the efficiency of combined IRE plus anti-PD-1 is associated with several factors, including DCs’ activation, the prompt release of danger-linked molecular patterns, and alleviation of the immunosuppressive environment of tumor [243].

Tumor-treating fields (TTFields) are a well-known noninvasive anticancer treatment technique that applies alternating electric fields of intermediate frequency (∼100–500 kHz) with limited intensity (1–3 V/cm) to interrupt the division process of cancer cells [244]. TTFields therapy is practically implemented for the treatment of patients with malignant pleural mesothelioma and glioblastoma multiforme. It has been reported that TTFields therapy could potentially impair microtubules’ polymerization and filaments of septin, which both are needed in the mitosis procedure for proper cytokinesis and segregation of chromosomes. Therefore, the application of this method could lead to mitotic catastrophe and consequent cancer cells’ death associated with aneuploidy [245,246]. Another study showed that TTFields therapy is able to induce the ICD process. The data of this study indicated that the combination of TTFields with anti-PD-1 for lung tumor treatment in the animal model led to a considerable reduction in tumor volume, in comparison with the alone treatment or control group. Furthermore, TTFields combined with anti-PD-1 therapy showed a remarkable increment in the infiltrating of CD45^+^ tumor cells [247]. PT-112, a new conjugate of platinum-pyrophosphate, is under development to be applied clinically for cancer treatment. In vitro studies reported that PT-112 could mediate the cytostatic and cytotoxic responses against numerous cell lines originated from human and mouse. The cytotoxic effects of PT-112 are related to the induction of danger signals initiating the anticancer immunity such as the exposure of calreticulin on the surface of dying cells and the release of HMGB1 and ATP molecules. It was demonstrated that the synergized administration of PT-112 and PD-1 or PD-L1 blockade in the animal model with immunologically competent settings could simultaneously induce immune effector cells and discharge immunosuppressive cells in the microenvironment of tumor [248].

N-(2-hydroxypropyl) methacrylamide (HPMA) copolymer with a degradable backbone has a significant advantage in targeting the facilitation of immunogenic drugs to improve their direct antitumor effects and ICD induction to provoke antitumor immunity. The copolymer, meanwhile, is able to mediate the crosslinking of PD-L1 on the surface according to its direction to the lysosome for degradation and also reveals persistent suppression. Furthermore, this two-pronged approach could employ and vivify slumbering T cells in the microenvironment of tumors and provoke the responses of T cells. KT-1 and MPPA, as a new framework for the application of polymer-based nanomedicines, can be used to reprogram immune responses in “cold” tumors and also strengthen the immune system to attack cancer cells. KT-1 could considerably induce ICD in vivo condition and also sensitizes cancer cells to checkpoint blockade through targeting of polymer-enhanced tumors [237].

Another study indicated that the encapsulated CRISPR/Cas9 in nanoparticles is able to knock out more specifically the function of cyclin-dependent kinase 5 gene to attenuate PD-L1 expression on cancer cells remarkably. More specifically, when paclitaxel (PTX), with a considerable capacity to convert “cold” tumor into “hot” tumor, encapsulated in noted nanoparticles it could remarkably reduce regulatory T lymphocytes, enhance antitumor immunity, induce immunogenic cell death, and repolarize tumor-associated macrophages [238].

Limagne et al. have shown that trifluridine/tipiracil and oxaliplatin when used in combination, ICD and antitumor CD8 activation, as well as depletion of Tumour-Associated Macrophages (TAM), in particular, TAM2, significantly were induced in MSS colorectal cancer [217]. Pfirschke et al., using lung adenocarcinoma mouse models, reported that applying oxaliplatin combined with cyclophosphamide in autochthonous tumors, without T cell infiltration and resisting common treatment choices, could effectively be sensitized to host antitumor T cell immunity through stimulating the innate immune system. Tumor cells were also sensitized to immune checkpoint inhibitors following stimulating tumor infiltration by T cells, and controlled cancer strongly [249].

Immunepotent CRP (ICRP), a mix of substances of low molecular weight gained from bovine spleens, has a cytotoxic effect on diverse tumor cell lines in vitro and regulates the immune response in vivo. Rodríguez-Salazar et al. have demonstrated that treatment murine melanoma B16F10 cells with the combination of ICRP and oxaliplatin enhanced the exposure and production of DAMPs, such as ATP and HMBG1. Interestingly, in vivo evaluations indicated that administration of tumor-derived DAMP-rich cell lysates obtained from B16F10 cells treated with the combination of ICRP and oxaliplatin inhibited melanoma tumor development and growth in a murine model [250]. Even more surprisingly, using immunostimulatory cytokines in order to potentiate ICD-inducing chemotherapy agents is another suggested approach. Accordingly, it has been postulated that treatment of colorectal cancer in a murine model with oxaliplatin in combination with IL-12 resulted in the eradication of metastatic colorectal tumors and inhibition of tumor recurrence as well as enhancing the ratio of CD8+ T lymphocyte/myeloid-derived suppressor cells (MDSCs) and CD8+ T lymphocyte/Tregs within the tumors [251]. These finding provided us a more comprehensive explanation that the combination therapy of tumors with antitumor agents that induce ICD is a great promise and potential therapeutic protocol.

### 6.2. Therapies Combining Radiotherapy-Induced ICD

Either for curative or palliative purposes, nearly half of cancer patients undergo ionizing irradiation either as a standalone intervention or in collaboration with chemo- and/or immunotherapeutic regimens [252]. The occurrence of DNA double-strand breaks is conventionally believed to be the root cause of the cytotoxic impacts of radiation on tumor cells. In line, DNA damage and subsequent tumor cell death has been attributed to four underlying principles (known as the 4 “Rs” of radiobiology): First, reorganization of cancer cells into radiosensitive phases of the cell cycle (G2/M); second, reoxygenation of formerly hypoxic cells within a TME; third, rectification of fatal DNA impairment and; the last, repopulation of living cancer cells, whereby the modifying of each factor changes cancer cell radiosensitivity. Nevertheless, the scope of the conventional perspective seems inadequate considering the whole context of the TME and host antitumor immunity [253]. Thus, ICD superseded the concept of cell death and encompasses the host’s immune system as a chief in-field responder to radiotherapy, which, in turn, instigates immune system and favorable systemic effects. Consequently, the fifth radiobiologic principle represents immune-mediated tumor rejection that is induced by radiation as an alternative radiosensitivity modality [254]. It is widely known that ICD is fostered by radiation therapy. Unlike the conventional forms of apoptosis, ionizing irradiation eliminates cancer cells but makes them emit a certain blend of signals that prompt the cross-prime between APCs and antigen-specific adaptive immune responses [44]. Indeed, radiotherapy with clinically relevant single doses reportedly stimulates the proliferation of APCs within tumor-draining lymph nodes (TDLNs), where antigen presentation and activation of CD8+ T cells occur. As a result, CTLs or NK cells drift back to the tumor site and kill the cancer cells by a mechanism in part mediated by perforin. Based on the yielded results, perforin deficiency in mice can inhibit the antitumor immunity, which was induced by X-rays [255]. Meanwhile, RT, the upshot of antitumor responses, can assist the immune system to regain its vigor by boosting the surface exposure of CRT and HSP, as well as HMGB1 release [116,256]. It was demonstrated that DC maturation and Interferon gamma (IFNg)-producing T cells get stimulated when tumor cells are irradiated both in vivo and in vitro. Murine models depict tumor antigen-specific T-cell responses when radiation-induced tumor cell death released HMGB1, which not only augmented the engulfment of antigenic materials by DCs via TRL4 but also mediated cross-presentation of tumor antigens into CD4 and CD8 T-cells [257]. It has been also indicated that membrane exposure of CRT prompted by means of radiation could heighten phagocytosis of dying cancer cells by DCs in vitro. However, murine models illustrated that both HMGB1 release and CRT cell surface expression are required for the accomplishment of antigen-specific T-cell response [234,258]. Elevated susceptibility of tumor cells to vaccine-mediated T-cell immune attack stems from the phenotype modification of target tissues induced by RT [259]. In vivo, RT can play the dual function of regulating the expression of the Fas Cell Surface Death Receptor (FAS) and MHC class I molecules on the tumor cells on one hand, and modulating MHC class I-mediated antitumor immunity on the other hand. The latter function is feasible by functionally affecting DCs’ antigen presentation. Also, peptide generation and the surface expression of MHC class I are amplified when X-ray treatment targets them and leads to the enlargement of their repertoire [260]. Additionally, RT determines the detectability of tumors by the immune system and immunogenics via modulating the phenotype of tumor cells. Despite the fact that T regulatory cells can diminish in numbers in the face of RT, they demonstrate impressive resistance to the cytotoxic effect of ionizing radiation. This is well supported by the fact that the antineoplastic effect of RT enhances when regulatory T-cells are thoroughly depleted [260,261]. It is interesting to note that, alongside X- or g-irradiation, vaccination with a-irradiated (bismuth-213) murine adenocarcinoma MC-38 also induces a lasting protective antitumor response in the murine model depending on tumor-specific T-cells. The 213-Bi can potentiate MC-38 cells to release DAMPs and stimulate dendritic cells in vitro. Even though insufficient evidence is available for ICD induction in human cancer patients receiving RT, the increasing emergence of clinical evidence has attracted considerable interest in investigating the mechanisms of IR-induced ICD [262]. In fact, most of the mechanisms and features of ICD have been defined with the use of chemotherapeutic compounds. Antitumor immune response in the TME is enhanced when IR and chemotherapeutic agents are used [263], deploying ICD pathways, which eventually set off the cross-presentation of tumor-derived antigens by DCs. However, in the clinical setting, the occurrence of tumor cell death as a prerequisite to initiate immune-mediated tumor rejection by each treatment alone may not achieve tumor cell death. Thus, the elicitation of ICD requires extensive studies to ascertain the optimum alternation between chemotherapeutic treatments and IR [264].

In order to upgrade the efficacy of therapeutic protocol and to prolong tumor immune response, using radiotherapy and immunotherapy as a combination therapy represents a potential chance. In several tumor models, including breast, lung, melanoma, and colon cancers, the potential systemic effects of the link between radiotherapy and checkpoint inhibitors have been established [265,266,267,268]. More recently it was revealed that in metastatic melanoma patients dual immune checkpoint blockade combinations with radiotherapy prolong survival and inhibit tumor progression. Moreover, a recent phase I trial in patients with metastatic melanoma evaluating ipilimumab (anti-CTLA4 antibody) in combination with radiotherapy was documented that only 18% of patients represented a partial response in un-irradiated groups. To investigate the cause of the difference in response, these experiments were repeated in the murine model, alike to the phase I clinical trial, obtaining only 17% of responses. However, use of anti-CTLA4 plus an anti-PD-L1 and radiation heightened CD8+/Treg ratio and showed a complete response rate of 80% [269].

Given that radiotherapy alters the TME and promotes releases tumor-derived antigens, Zegers et al. conducted a study which hypothesized that radiotherapy in combination with L19-IL2 offers an improved antitumor effect. They reported that combining L19-IL2 with radiotherapy meaningfully increased the CTL population in murine models of cancer and stimulated cytotoxic CD8+ T lymphocyte responses [270]. Another preclinical study, in a murine model of lung cancer, also demonstrated that radiotherapy plus an adjuvant agonist antibody against CD134 (OX40) resulted in improved antitumor immune responses and tumor control [271]. In line with this study, an animal model study indicated that radiotherapy combined with anti-CD40 antibody therapy induced a CD8+ T cell-mediated immunity against B-cell lymphoma [272]. On the other hand, Deng et al. suggested that anti-PD-L1, when administrated in combination with radiotherapy, can decrease the numbers of MDSCs and, therefore, their suppressive effects on the tumor immune system [267].

### 6.3. Therapies Combining Photodynamic Therapy-Induced ICD

Photodynamic therapy (PDT), an FDA-approved clinical protocol, is the latest therapeutic modality presented by a renowned association with certain DAMPs which are used for various malignant and nonmalignant diseases. The traditional phototherapy turned to modern photodynamic therapy as a result of experiments on the combination of light and chemicals for better therapeutic efficiency. Using a chemical component that acts as a photosensitizer indicates that PDT is considered a physicochemical, rather than an exclusively physical, antitumor modality. Minimal invasion, low mutagenic potential, low systemic toxicity, and pinpoint tumor targeting make PDT have an edge over classical anticancer factors like surgery, ionizing radiation, and chemotherapy [273]. In the face of photodynamic injury, cancer cells either resort to inaugurating a rescue-response strategy or surrendering themselves to multiple cell deaths. Following PDT, three ways are involved in reducing and/or eliminating tumors: ROS production that causes cancer cell death, induction of ischemia in TME through the elimination of tumor-related vasculature that deprives cancer cells of oxygen and nutrients, and finally by recruitment of inflammatory and immune factors promoting tumor elimination and cancer cells’ recognition by infiltrated leukocytes [10]. The production of ROS and oxidative stress-originated PDT is the outcome of a two-step process including the exploitation of tumor-localizing photosensitizer and subsequently its stimulation by absorbing a particular wavelength of light. The propensity of a specific photosensitizer to depict a certain rate of tropism toward a particular subcellular organelle or locale has potentiated PDT for a precise direction of oxidative stress [274]. Studies during the late 1980s and early 1990s also demonstrated the recruitment of inflammatory cells like macrophages, leukocytes, and lymphocytes into the tissues treated with PDT [207,208], indicating induction of the immune system. Increased production of IL-6 and IL-1, but not TNF-α, have also been reported as a result of photodynamic therapy [209].

Effective photosensitization mainly relies on the Photosensitizer (PS) physicochemical features, comprising selectivity for cancer cells, chemical purity, chemical and physical durability, buildup within tumor cells as soon as drug administration, stimulation at a wavelength with optimum tissue dissemination, and quick elimination from normal tissues [275]. If mitochondria hosts a photosensitizer, mitochondrial oxidative stress pursues and, thanks to the fact that various subcellular localizations can house a wide range of photosensitizers, ICD concept can be investigated from different angles. A bright outlook is conceived for ICD induction considering the unique trait of PDT treatment with ER-localizing synthesizers due to the existing link between immunogenic apoptosis and ER stress [10,276].

Although PDT, either individually or in conjunction with other therapeutic modalities, has been the main focus of over 200 clinical trials since initial work in the 1970s, scant information has been gained regarding its implications for the human immune systems. Not only can systemic antigen-specific immune responses be intensified but also distant, nontreated tumors can be affected by the induction of clinical abscopal effect, like immune response [12]. Although the impressive immune-modulatory potential of PDT makes it apt to devise effective cancer disease-management plots, high-inflammatory PDT factors can trigger acute inflammation typified by heightened expression of pro-inflammatory cytokines, adhesion molecules’ E-selection and Intercellular Adhesion Molecule 1 (ICAM-1), and quickly amass leukocytes in the treated tumor area [277]. PDT improvement of antitumor immunity seems to be concomitant with the activation of DCs sparked by dying cells. Definitely, DC maturation and activation, as well as T cell activation, are induced following the incubation of photosensitized tumor cells with immature DCs [278]. Intensified immunological detection of tumor-associated antigens (TAAs) by PDT was first revealed in clinical experiments; more exactly, it was detected that 5-aminolevulinic acid (5-ALA)- and porfimer sodium (Photofrin)-based PDT can heighten the host immune reaction against basal cell carcinoma-associated tumor antigen and hedgehog interacting protein 1 (Hip-1) [278]. DC maturation and IL-12 expression are attributed to lysate secreted by PDT. Even PDT-derived vaccines can induce both innate and adaptive immune system [279]. HSPs’ proteins and HSP70, in particular, are the perfectly characterized DAMPs induced by PDT-triggered cell death with the capacity for immunogenicity induction, which evokes tumor-associated macrophages (TAMs) to generate complement proteins when mediated by TLR2 and 4 receptors alongside NF-kB activation [280].

A recent study have proved hypericin-based PTD (Hyp-PTD) to be the ideal ICD inducer both in murine and human systems, promising perfect preclinical and clinical exploitation [116]. ICD activated by Hyp-PDT was much more operative when compared to that stimulated by chemotherapy or radiotherapy. Principally, using Hyp-PDT prevents tumorigenesis by impeding tumor-promoting cytokines’ signaling and downregulating mediators of tumor metastasis like matrix metalloproteinase-9 (MMP9). It has been reported that Hyp-PDT strongly inhibits transcriptional functions of NF-kB and AP-1 in treated cancer cells. Moreover, Hyp-PDT can reduce the cancer cell-secreted tumor-promoting cytokines, such as GM-CSF, IL-6, and TNF [175]. Although there have been many clinical trials of PDT treatment in various cancers [12], more research is needed to determine all the functional features of Hyp-PDT. Thanks to the observed engagement of CRT and ATP in cell death apoptosis, having resulted from photodynamic sensitization, Agostinis’s laboratory divulged that ICD can be elicited when CRT exposure and ATP secretion occur simultaneously during PDT [281,282].

Taken together, by inducing ICD, PDT can withstand cancer recurrence via educating the immune system. Potentiating this characteristic by stimulating several components of the immune system has been of great interest. This feature can be achieved by using PDT in combination with wide-ranging immunostimulant agents. Since PDT also modulates the cell membrane penetrability, adding it as an adjuvant to chemotherapy may raise the deliverability of cytotoxic drugs [283]. One of the most interesting aspects of using PDT in combination with chemotherapy is the enhanced treatment efficiency of human uterine sarcoma cell lines and drug-resistant murine leukemia. It was demonstrated that in vitro and in vivo antitumor immune response effect significantly improved when cisplatin was administrated plus PDT, considering that cisplatin does not have an immunomodulatory effect [284,285]. In combination with chemotherapy, some other therapies have been developed. One of the most remarkable findings was the increased efficacy of therapy-resistant murine leukemia and human uterine sarcoma when PDT was added. PDT might verify a hopeful addition to chemotherapy. It was also revealed that PDT in combination with cisplatin leads to a significantly augmented in vitro and in vivo antitumor result that can demonstrate value when considering the toxicity of cisplatin treatment.

It has long been reported that Tregs produce immunosuppressive cytokine-like TGF-β that inhibit DCs’ activation. Castano et al. have shown that Tregs can be reduced via using cyclophosphamide in combination PDT, which resulted in long-lasting J774 reticulum cell sarcoma cure and inhibited the tumor recurrence [286]. As noted before, CRT expressed on the surface of tumor cells following treatment with numerous cancer therapies like PDT. In this matter, Korbelik et al. evaluated the effect of exogenously added CRT for improving the antitumor effect mediated by PDT [277]. They showed that antitumor immune response, prompted via PDT or PDT-generated vaccines, could be enhanced by externally added CRT and can be an operative adjuvant for cancer therapy with PDT [287].

Doix et al. indicated that the use of PDT-based DC vaccination before radiotherapy fails to prevent and control tumor growth, whereas PDT-based DC vaccination protocol in the peri-radiotherapy period could exhibit tumor growth inhibition, highlighting the significance of the coincidence of T cell activation and adjustments of the tumor microenvironment [288]. In another study, it was documented that a combination of photosensitizer pheophorbide A (PheoA), ROS-responsive paclitaxel dimer prodrug (PXTK), and anti-PD-L1 peptide dPPA exhibited significant antitumor and antimetastasis effect, emphasizing the value of the combination of chemotherapy, PDT, and immunotherapy in cancer management [289].

## 7. Conclusions

Cancers are serious diseases causing death, and novel therapeutics with developed efficacy and safety are necessary. Immunotherapy is deliberated as one of the hopeful therapeutic strategies for curing cancer. Currently, cancer cells’ immunogenicity has been recognized as a vital factor in efficient cancer therapy. Researchers have recognized that besides therapeutically exploiting innate or adaptive immune cells directly or improving the functions of T cells, cancer cells also need to be made immunogenic. According to studies, stimulating antitumor immunity, besides the tumor cells’ death, can result in durable therapeutic benefits. This may be understood by stressing or damaging tumor cells, accompanied by the emission of the signals called DAMPs. These signals act via receptors on dendritic cells to induce the tumor antigens’ presentation to T cells. Numerous drugs have been recognized that are able to stimulate ICD when involved as therapeutic factors. These include several chemotherapeutics applied in the clinic as well as several anticancer agents and combinational therapies that are still under preclinical or clinical development. It was revealed that the stimulation of adaptive immunity against dying cancer cells is related to improved disease outcome in patients with different cancers. Therefore, new therapeutic factors that stimulate ICD are immediately expected.

## Figures and Tables

**Figure 1 cancers-12-01047-f001:**
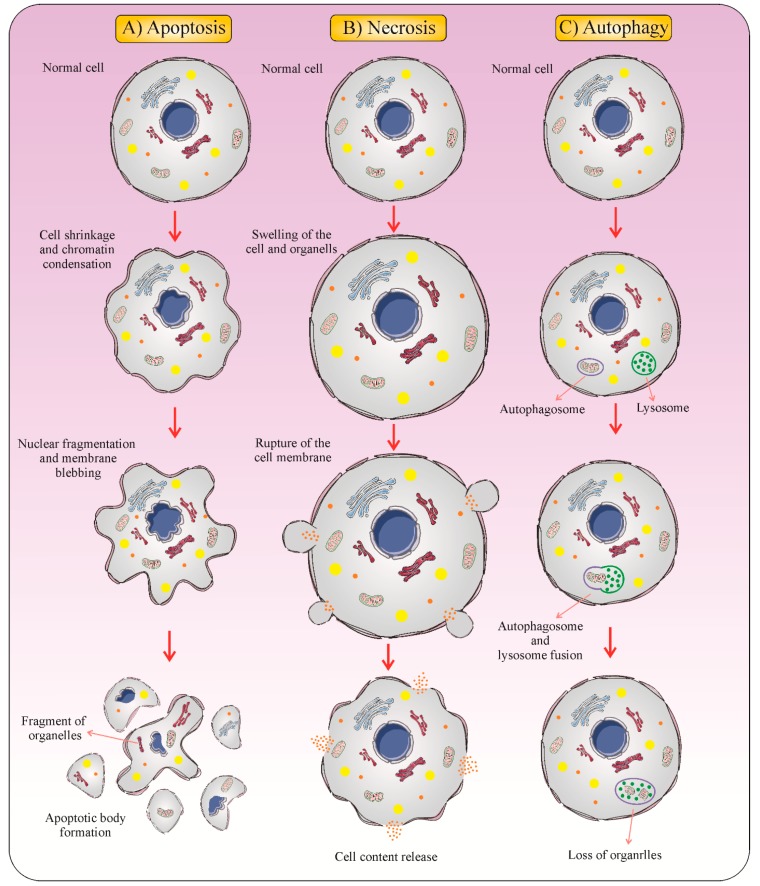
Three different forms of cell death. (**A**) Apoptosis is a form of cell death that some molecular mechanisms in a cell lead to its death. It is generally characterized by distinct morphological features including cell shrinkage, blebbing, and nuclear fragmentation. (**B**) Necrosis is a type of cell death that is morphologically described by an increasing translucent cytoplasm, swelling of organelles, and increased cell volume. (**C**) Autophagy is the major intracellular degradation process by which cytoplasm ingredients are translocated to and degraded in the lysosome.

**Figure 2 cancers-12-01047-f002:**
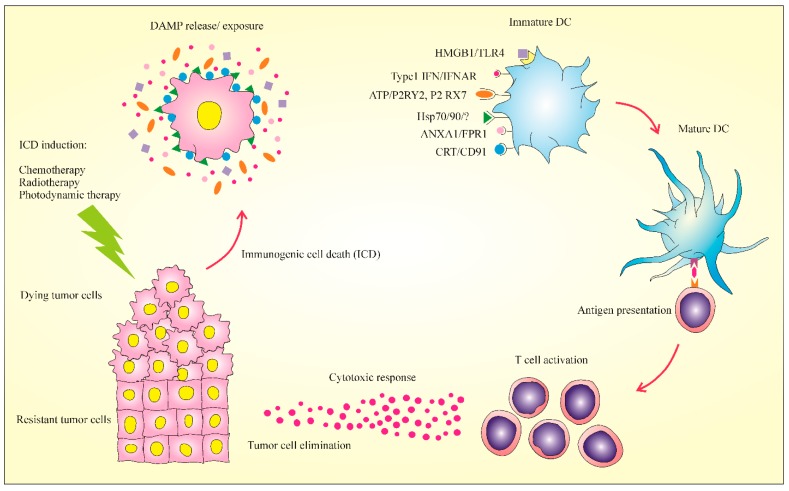
Major hallmarks of immunogenic cell death: The use of ICD inducers alone or in combination with other anticancer therapies stimulates the exposure or release of DAMP molecules. These DAMP molecules released from dying tumor cells stimulate the DCs’ recruitment into the tumor and activate T cells. Abbreviations: ICD; Immunogenic cell death, DAMP; danger-associated molecular patterns.

**Figure 3 cancers-12-01047-f003:**
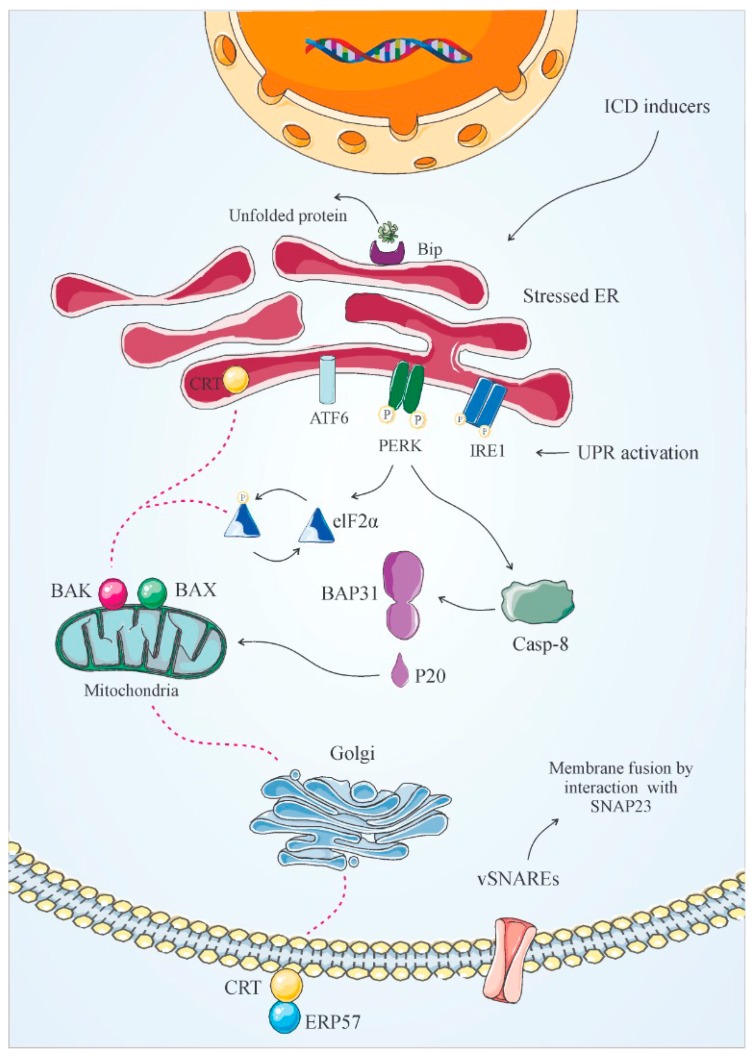
Mechanism of CRT exposure: Several steps are involved in the exposure of CRT in response to immunogenic cell death inducers including activation of PERK, the phosphorylation of eIF2α, caspase-8-dependent cleavage of BAP31, and the activation of Bax and Bak. Finally, a pool of CRT that has transited the Golgi apparatus is secreted by SNARE-dependent exocytosis (see text for detail). Abbreviations: CRT; calreticulin, PERK; protein kinase R-like ER kinase, BAP31; B cell receptor–associated protein 31, Bax; BCL2-associated X protein, Bak; Bcl-2 homologous antagonist/killer, SNARE; soluble N-ethylmaleimide-sensitive fusion protein-attachment protein receptor.

**Table 1 cancers-12-01047-t001:** Type I and type II inducers of immunogenic cell death.

ICD Inducers	Molecular Mechanism(s) of Action	Mechanism(s) of ICD Induction	Ref.
Type I			
Cardiac glycosides	Targeting Na+/K+ pump, upregulation of DR-4 and -5, TIAM1 and PAR4, inhibition of DNA topoisomerase	CRT exposure, ATP secretion and passive release of HMGB1	[117]
Capsaicin (CPS)	TRPV1-mediated signaling	Ectopic expression of CRT and HSP90, release of ATP, HMGB1, HSP70 and HSP90	[118]
Clostridium difficile toxin B	Modification of Rho protein activity	CRT exposure, secretion of ATP, HSP70, HSP90 and HMGB1 release and autophagy	[119]
High hydrostatic pressure (HHP)	-	CRT, HSP70 and HSP90 exposure, ATP release, increased expression of maturation markers on DCs, increased cytokine production in DC	[120]
Ultraviolet light (UV) C	DNA damage	CRT exposure, HSP70 and HMGB1 release, strong phagocytosis activity of DCs, proinflammatory cytokine production in DCs.	[52,121]
Type II			
Pt-N-heterocyclic carbene (NHC)	ER associated production of ROS	Surface-exposed CRT and the extracellular release of ATP and HMGB1	[122]
Coxsackievirus B3	Accumulation of viral proteins in the ER and ER stress	Increasing the CD8+ lymphocytes, development of tumor-nourishing blood vessels	[123,124]

Abbreviations: ICD, immunogenic cell death; CRT, calreticulin; DAMP, danger-associated molecular patterns; ER, endoplasmic reticulum; DR-4, death receptor-4; TRPV-1, transient receptor potential vanilloid type-1; DC, dendritic cell, ROS, reactive oxygen species.

**Table 2 cancers-12-01047-t002:** ICD and combination therapy in cancer.

Therapeutic Agents	Mechanism	Cancer Type
Therapies combining chemotherapy-induced ICD		
Doxorubicin plus IL-18	increased expression of MHC class I and Fas by ID8 murine ovarian cancer cells, sensitization to CTL and Fas-mediated killing	ovarian cancer
Doxorubicin plus DCs	increased frequency of CD8+ T cells, serum interferon-γ levels	osteosarcoma tumors
5-fluorouracil plus folinic acid, oxaliplatin	induction of ICD suppressed the expression of PD-L2	colorectal cancer
Trifluridine/tipiracil plus oxaliplatin	stimulation of ICD and antitumor CD8 cells, depletion of TAM	colorectal cancer
LTX-401 plus double checkpoint inhibition of PD-1 and CTLA-4	reduce the growth of the tumor	fibrosarcomas
Lurbinectedin plus CTLA-4/PD-1 dual checkpoint blockade	extended life expectancy, tumor clearance	osteosarcoma
IRE plus anti-PD-1	activation of DCs, alleviation of immunosuppressive tumor microenvironment	PDAC
TTFields plus the anti-PD-1	decrease in tumor volume, increases in CD45^+^ tumor infiltrating cells	Lung cancer
CRISPR/Cas9-Mediated Knockout of the Cdk5 Gene plus paclitaxel	reduce regulatory T lymphocytes, repolarize tumor-associated macrophages, enhance antitumor immunity	colorectal cancer cells (CT26), murine melanoma cells (B16F10) and murine fibrosarcoma
ICRP plus Oxaliplatin	inhibit melanoma tumor development and growth	melanoma
oxaliplatin plus IL-12	inhibit tumor recurrence, enhance the ratio of CD8 + T lymphocyte /MDSCs and CD8+ T lymphocyte /Tregs within the tumors	colorectal cancer
Therapies combining radiotherapy-induced ICD		
CTLA-4/PD-1 dual checkpoint blockade plus radiotherapy	heighten CD8^+^/Treg ratio, complete response rate of eighty percent	metastatic melanoma
L19-IL2 plus radiotherapy	increases in the CTL population in murine models of cancer stimulation of cytotoxic CD8+ T lymphocyte responses	CT26 colon tumors
Radiotherapy plus an adjuvant agonist antibody against CD134 (OX40)	improved antitumor immune responses and tumor control	lung cancer
Anti-CD40 plus radiotherapy antibody	stimulate CD8+ T-cell mediated immunity	B-cell lymphoma
Therapies combining photodynamic therapy-induced ICD		
Cisplatin plus PDT	improved cancer cell death, diminishing the toxicity of antineoplastic drugs	cervical cancer
Cyclophosphamide plus PDT	decrease Tregs population, inhibit the tumor recurrence	reticulum cell sarcoma
PDT-based DC vaccination plus radiotherapy	significant tumor growth delay	squamous cell carcinoma
pheophorbide A plus PXTK and dPPA	activation of CD4+, CD8+ T cells and NK cells, enhances secretion of cytokines (TNF-α and IL-12), tumor growth inhibition	breast cancer

Abbreviations: PDAC, pancreatic ductal adenocarcinoma; DC, dendritic cell; dPPA, anti-PD-L1 peptide; PXTK, paclitaxel dimer prodrug; PD-1, programmed cell death protein 1; CTLA-4, cytotoxic T lymphocyte-associated protein 4; IRE, irreversible electroporation; ICRP, immunepotent CRP; Cdk5, cyclin-dependent kinase 5; MDSC, myeloid-derived suppressor cells; CTL, cytotoxic T lymphocytes; PDT, photodynamic therapy.

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
