# Peer review of "Current Approaches for Combination Therapy of Cancer: The Role of Immunogenic Cell Death"

_cancers, 2020, doi:10.3390/cancers12041047_

Round 1

Reviewer 1 Report

Manuscript ID: cancers-757784

Title: An emerging approach for combination therapy of cancer: the role of immunogenic cell death

Page1, Line 38-41: This statement should be re-formulated. The sentence is grammatically un-correct, not clear and too long.

Page2, Line 73: The definition of autophagy as a cell death mechanism or cell survival mechanism is not clear and simplified too much. The authors should describe better and in-depth.

Page2, Line 93: It is not clear how necrotic cell products did induce immunological cell death. An explanation of the relevant mechanism should be addressed.

Page3, Line 94: The distinction between immunogenic versus tolerogenic cell death requires a better explanation.

In section 3, the authors indicate how each biochemical hallmark molecule can induce anti-tumour immunity. Therefore given the anti-tumour immunity topic, the authors should describe and define the overall immunity against cancer, which is missing in the paper.

The following sections (4. ICD inducers) are organized in a somewhat confusing manner. In particular, the distinction between ER-stress and UPR is again not obvious to the average reader and requires a better explanation and definition.

Page 7, Line 269: Please define  “physical modalities” referred to type Ⅱ inducers.

In section 6, the authors introduce combinational treatments with PD-1 blockade approaches (Page 14, Line 575). The authors should describe and define immune-checkpoint inhibitors.

Finally, most of the references cited appear as out-dated.  Although normally, the authors refer to the original experimental manuscript when appropriate, in a review article related to such a fast-moving field, they should refer to more recent articles (only 4 cited were published in 2018 and 1 cited were published in 2019 over 231).

Author Response

Thank you very much for the valuable comments provided. We have addressed all the comments as shown in the revised manuscript and detailed below. We have restructured the paragraphs, added more text, references, and added a new figure (Fig. 3) and table (Table 2). We also updated the title of the manuscript as the editor mentioned. All of the changes in the manuscript are highlighted in yellow.

Page1, Line 38-41: This statement should be re-formulated. The sentence is grammatically un-correct, not clear and too long.

We apologize for this error in the initial manuscript, it has been corrected in the revised manuscript.

Immunotherapy has been considered a promising therapeutic modality in oncology that aims to re-activate the immune system that is stopped by the tumor cells and creates a robust anti-tumor response.

Page2, Line 73: The definition of autophagy as a cell death mechanism or cell survival mechanism is not clear and simplified too much. The authors should describe better and in-depth.

We want to appreciate this point; we have mentioned in the revised version.

Type 2 cell death or autophagy, is an intracellular degradation process that plays a central role in the protection of cells and organisms from stressors. Autophagy is described by the removal of materials marked for destruction into double-membrane vesicles called autophagosomes. Autophagosomes fuse with endosomes and then they deliver their content to the hydrolytic interior of lysosomes for degradation. Autophagic degradation is considered as a vital source of amino acids, nucleotides and fatty acids and provides energy for cells. Autophagy plays a part in physiological processes and its activity is fundamental for adjustment to starvation, cell development, aberrant structures degradation, cell survival, homeostasis, and regulation of cell death.

Page2, Line 93: It is not clear how necrotic cell products did induce immunological cell death. An explanation of the relevant mechanism should be addressed.

During revision, a general review was conducted and the mentioned points were done in the current version of the manuscript.

Unlike apoptosis, a regulated cell death process in which the plasma membrane does not rupture, necrosis is inevitably associated with the release of DAMPs and intracellular organelles. Pathophysiologically, this stimulates an immune response because new surfaces become available to both the innate and the adaptive immune system during a process defined as necroinflammation.

Page3, Line 94: The distinction between immunogenic versus tolerogenic cell death requires a better explanation.

We appreciate this point too and have discussed related the issue in the new revision of the manuscript.

ICD describes a functionally distinct type of cell death that results in a T cell-mediated immune response, particularly for dead cell-derived antigens. ICD can be stimulated by different mechanisms and leads to translocation or release of DAMPs from the dying cells that activate the immune response. In contrast to ICD, tolerogenic cell death (TCD) including most types of apoptosis is a non-inflammatory mechanism of cell death that is described by membrane blebbing and loss of DAMP secretion. TCD is unable to stimulate an immune response associated with cell death.

In section 3, the authors indicate how each biochemical hallmark molecule can induce anti-tumour immunity. Therefore, given the anti-tumour immunity topic, the authors should describe and define the overall immunity against cancer, which is missing in the paper.

We appreciated your suggestion we have revised in the current version.

Naturally, the immune system can recognize and destroy cancer cells and play a significant role in the regulation of tumor progression. The immune system is educated in such a way that it does not respond to normal cells. While, several mutations in cancer cells result in the expression of tumor-specific antigens that can be identified as non-self and activate the immune system, finally resulting in the elimination of cancer cells. The term “antitumor immunity” defines the innate and adaptive immune responses that regulate tumor. Both innate and adaptive immunity play a role in the identification and fighting against tumors and a successful antitumor immune response is related to the close interaction of several factors of innate and adaptive immune responses. They are composed of antigen-presenting cells, various subsets of T cells, B cells and NK cells. However, tumors use several ways of immunosuppression to stop the antitumor effect of immune cells. Dysregulation of the balance between the effector and regulatory cell compartments is one of the key strategies for tumors to escape immune eradication. A better understanding of the vital immune cells and the regulatory networks participated in the interaction between tumor cells and the immune system is central for the improvement of therapeutic strategies to strengthen the immune system against cancers.

The following sections (4. ICD inducers) are organized in a somewhat confusing manner.

Thanks for the mentioned points, during the revision processes, we revised the mentioned point and added a new part as a Photodynamic therapy (PDT).

Phototherapy or the use of light to cure disease has been established in ancient Indian, Egyptian and Chinese civilizations. The traditional phototherapy turned to modern photodynamic therapy (PDT) as a result of experiments on the combination of light and chemicals for better therapeutic efficiency. Using a chemical component that acts as a photosensitizer indicates that PDT is considered a physicochemical rather than an exclusively physical anti-tumor modality. Exposure of cells to PDT starts a chain reaction leading to the production of ROS and ultimately cell death caused by oxidative stress. Studies during the late 1980s and early 1990s also demonstrated the recruitment of inflammatory cells such as macrophages, leukocytes, and lymphocytes into the tissues treated with PDT, indicating activation of the immune system. Up-regulation of pro-inflammatory cytokines, IL-6 and IL-1 but not TNF-α, have also been reported as a result of photodynamic therapy. Comparing tumor growth in immunodeficient and immunocompetent Balb/C mice showed that long-term effects of PDT therapy were attributed to the activation of the anti-tumor immune response as tumor development was frequently occurred in deficient but not in normal mice, whereas short-term responses were similar in both normal and immunocompromised mice. The anti-cancer properties of PDT are relevant to direct impacts on cancer cell death and its immunogenic features. PDT exerts cytotoxic activity against malignant cells through activating anti-tumor immunity in general and by inducing bona fide ICD. Unlike chemotherapy, very little is known about the molecular mechanisms that account for the immunostimulatory functions of PDT. It has been identified that the immunogenic demise driven by PDT exposure is accompanied by CALR, HSP70 and HSP90 translocation from ER to the plasma membrane and extracellular secretion of HMGB1 and ATP. PDT based on the photosensitizer hypericin preferentially targets the ER and induces ROS formation adjacent to the ER. Hypericin predominantly causes ER-associated oxidative stress and initiates different signaling pathways of UPR which finally leads to amplified mitochondria-dependent apoptosis. Kinetically, the emission of DAMPs (ecto-CRT, HSP70 and secreted ATP) in response to hypericin-based PDT did occur more quickly than those previously reported for these DAMPs. Importantly, Hyp-PDT has several advantages over classical anti-cancer regimens. ICD induced by Hyp-PDT was much more effective when compared to that induced by chemotherapy or radiotherapy. Specifically, using Hyp-PDT prevents tumorigenesis by impeding tumor-promoting cytokines signaling and down-regulating mediators of tumor metastasis like cancer-derived matrix metalloproteinase-9 (MMP-9). It has been reported that Hyp-PDT strongly inhibits transcriptional activities of NF-kB and AP-1 in treated cancer cells. Moreover, Hyp-PDT can reduce the cancer cell-secreted tumor-promoting cytokines such as GM-CSF, IL-6, and TNF. Although there have been many clinical trials of PDT treatment in various cancers, more research is needed to determine all the functional features of Hyp-PDT.

In particular, the distinction between ER-stress and UPR is again not obvious to the average reader and requires a better explanation and definition.

We would like to thank for your careful review, we have explained more in the revised paper. Being a cell factory for protein production and assembly, the ER network possesses a precise and rigorous quality control system called (ERQC) for final monitoring of the synthesized biomolecules. When subjected to biotic and abiotic environmental stressors, the levels of aberrant unfolded/misfolded proteins may overwhelm the quality control systems thus allowing protein aggregates to be formed in the ER. The resulting imbalance between the cellular demand for and capacity of ERQC is referred to as ER stress. When ER stress is prolonged or intense, it can cause ER and intracellular organelle damage or cell death in extreme cases. Accumulation of aberrant proteins triggers the unfolded protein response (UPR) which in turn potentiates ERQC activities to reduce the burden of misfolded and unfolded proteins entering the ER.

Page 7, Line 269: Please define “physical modalities” referred to type Ⅱ inducers.

We have added a new part to the definition of the issue. Physical cell death-inducing modalities such as Photodynamic Therapy (PDT) have been confirmed to be able to act as in situ vaccines, and to aid in inducing antitumor immunity in human patients; PDT most probably by inducing ICD.

In section 6, the authors introduce combinational treatments with PD-1 blockade approaches (Page 14, Line 575). The authors should describe and define immune-checkpoint inhibitors.

Thanks for the comments and suggestions we have added to the current version.

After cessation of the chemotherapy process, residual tumor cells are able to upregulate adaptively the programmed death ligand-1 (PD-L1) so as to interact with the receptor of the immune checkpoint on the surface of T cells, which is a kind of escape system from immunosurveillance (immunosuppressive state). Therefore, there is a significant challenge in immunotherapy applications in cancers with immunosuppressive state [237]. It has been verified that monoclonal antibodies application for blocking of immune checkpoints could potentially provide high clinical achievement for cancer immunotherapy; however, their application is limited due to the cold environment of tumors [238]. Recently, many attentions have been focused to develop new and effective inhibitors for immune checkpoints in the immuno-oncology research area. Tumor cells could potentially escape from immunosurveillance and proceed with their specific mechanisms such as the pathway activation of immune checkpoints to repress the anti-tumor responses of the immune system. The inhibitors of immune checkpoints are able to revitalize the anti-tumor responses of the immune system through interfering with inhibitory effects on signaling pathways and also enhance the elimination of tumor cells by immune-mediated actions [239]. The implementation of such inhibitors against the PD-1/PD-L1 axis and probably CTLA-4 (cytotoxic T-lymphocyte antigen-4) has indicated considerable clinical benefits and even transformed rapidly as a practical method in medical oncology. Application of monoclonal antibodies, as potential co-inhibitors for immune checkpoints particularly CTLA-4 and PD-1, has been utilized clinically for various malignancies, including bladder cancer, head and neck squamous cell carcinoma, Hodgkin lymphoma, non-small cell lung cancer, Merkel cell carcinoma, melanoma, MSI-high colorectal carcinoma and renal cell carcinoma [240]. These inhibitors also have improved the possible practical methods for cancer treatments in the oncology field. Ipilimumab, as the first confirmed inhibitor for immune checkpoints that targets CTLA-4 molecules, has been applied for the treatment of those patients who suffer from advanced melanoma. This inhibitor is able to prevent the inhibition of T-cells and also promotes the proliferation and activation of T cells’ effectors. The other inhibitors, nivolumab, and pembrolizumab that target PD-1 indicated promising results in the treatment of NSCLC (non-small cell lung carcinoma) and melanoma with the rate of objective response between 40 and 45%. Furthermore, the treatment of urothelial bladder cancer with inhibitors of PD-1/PD-L1 pathways revealed an increment in response rate in the range of 13 and 24% [239].

Different clinical experiments have recently demonstrated that the use of anthracyclines as an ICD inducer led to an increase in the blocking of PD-1 and PD-L1 efficacy. These data could support the beneficial concept of ICD in cancer treatment [241]. LTX-401 is a known conjugated antibody- T-PNU drug, crizotinib as an inhibitor for tyrosine kinase and oxaliplatin as a chemotherapeutic agent, have a natural potential to induce ICD. It was reported that the combination of  LTX-401 and the inhibitors of CTLA-4 and PD-1  indicated considerable antineoplastic effects on primary distant and lesions tumors, showing the higher therapeutic effect of ICD and checkpoint blocker with together [241]. A study showed that lurbinectedin treatment could induce ICD traits such as calreticulin exposure, INF-I immune responses, ATP release and the exodus of HMGB1 molecules at in vitro condition.  The anti-cancer and antineoplastic properties of lurbinectedin could be strengthened remarkably by its combination with the double blockade of CTLA-4 and PD-1 immune checkpoints [242]. In another study conducted by Zao et al., it was revealed that irreversible electroporation (IRE) combined with anti-PD-1 could significantly induce the ICD process and also mediates the consequence durable response in a treated model (orthotopic PDAC). The mechanistic investigations disclosed that the efficiency of combined IRE plus anti-PD-1 is associated with several factors, including DCs activation, the prompt release of danger linked molecular patterns and alleviation of the immunosuppressive environment of tumor [243].

Tumor-treating fields (TTFields) are a well-known noninvasive anticancer treatment technique that applies alternating electric fields of intermediate frequency (∼100-500 kHz) with limited intensity (1-3 V/cm) to interrupt the division process of cancer cells [244]. TTFields therapy is practically implemented for the treatment of patients with malignant pleural mesothelioma and glioblastoma multiforme. It has been reported that TTFields therapy could potentially impair microtubules’ polymerization and filaments of septin, which both are needed mitosis procedure for proper cytokinesis and segregation of chromosomes. Therefore, the application of this method could lead to mitotic catastrophe and consequence cancer cells death associated with aneuploidy [245, 246]. Another study showed that TTFields therapy is able to induce the ICD process. The data of this study indicated that the combination of TTFields with anti-PD-1 for lung tumor treatment in the animal model led to a considerable reduction in tumor volume in comparison with the alone treatment or control group. Furthermore, TTFields combined anti-PD-1 therapy showed a remarkable increment in the infiltrating of CD45+ tumor cells [247]. PT-112, a new conjugate of platinum-pyrophosphate, is under development to be applied clinically for cancer treatment. In vitro studies reported that PT-112 could mediate the cytostatic and cytotoxic responses against numerous cell lines originated from human and mouse. The cytotoxic effects of PT-112 are related to the induction of danger signals initiating the anticancer immunity such as the exposure of calreticulin on the surface of dying cells and the release of HMGB1 and ATP molecules. It is demonstrated that the synergized administration of PT-112 and PD-1 or PD-L1 blockade in the animal model with immunologically competent settings could simultaneously induce immune effector cells and discharge immunosuppressive cells in the microenvironment of tumor [248].

HPMA copolymer with a degradable backbone has a significant advantage in targeting the facilitation of immunogenic drugs to improve their direct antitumor effects and ICD induction to provoke antitumor immunity. The copolymer, meanwhile, is able to mediate the crosslinking of PD-L1 on the surface according to its direction to the lysosome for degradation and also reveals persistent suppression. Furthermore, this two-pronged approach could employ and vivifies slumbering T cells in the microenvironment of tumors and provoke the responses of T cells. KT-1 and MPPA, as a new framework for the application of polymer-based nanomedicines, can be used to reprogram immune responses in “cold” tumors and also strengthen the immune system to attack cancer cells. KT-1 could considerably induce ICD in vivo condition and also sensitizes cancer cells to checkpoint blockade through targeting of polymer-enhanced tumors [237].  

Another study indicated that the encapsulated CRISPR/Cas9 in nanoparticles is able to knock out more specifically the function of cyclin-dependent kinase 5 gene to attenuate PD-L1 expression on cancer cells remarkably. More specifically, when paclitaxel (PTX) with a considerable capacity to convert “cold” tumor into “hot” tumor, encapsulated in noted nanoparticles could remarkably reduce regulatory T lymphocytes, enhance antitumor immunity, induce immunogenic cell death and repolarize tumor-associated macrophages [238].

Reviewer 2 Report

The review on ICD is a comprehensive but the following are suggestions to improve.

1) Add information or discussion on the dose used for different ICD inducers.  This is very relevant especially when using combination therapies.  Sometimes the dose of a drug that inducse ICD can have toxic side effects.

2) A figure detailing the pathway for CRT exposure would be helpful as well as better explanation of the role of ER stress and the UPR.  These are minimally discussed.

3) The title of the review does not really reflect the content since the discussion of the combination therapies is almost at the end and most of the review is general background on ICD. Recommending reducing this background content to what is relevant to introduce section 6.

4) Other than section 6 - which is the novel and critical part of the review, the information presented is old (many references are decades old) and reported in other similar reviews.  In fact there are only a couple of references from 2018 and one from 2019.  So much of the material is dated.

5) Addition of a table or figure for section 6 would bolster this information.  It is highly suggested to look at recent publications in 2019 and 2020 on ICD and combination therapy approaches, for example targeting the PD1-PD-L1 pathway to update the review.

Author Response

Thank you for this point. We did several modifications to the initial manuscript based on that your suggestions, in the revised version of the manuscript, we added new sections and updated most of the references. we hope the newer version shows more focus on the goal.

1) Add information or discussion on the dose used for different ICD inducers.  This is very relevant especially when using combination therapies.  Sometimes the dose of a drug that induce ICD can have toxic side effects.

Thank you very much for the mentioned point. This issue has been added to the revised version of the manuscript. We have added the dose which used for different ICD inducers.

  • The beneficial effects of cardiac glycosides (at a dose of 0.5 mg/kg) have also been associated with anti-tumor immunity
  • In the study of the effects of CPS in anti-tumor immunity, human bladder cancer cell lines undergoing apoptosis (dose-dependent apoptosis reaching approximately 30 % apoptotic cells using 150 μM CPS) were shown to emit ICD hallmarks after treatment with CPS in vitro.
  • It was shown that toxin-treated (500 ng/ml of TcdB) mouse colorectal cancer CT26 cells were immunogenic and caused mouse bone marrow-derived dendritic cell activation in vitro
  • In the study of the effects of CPS in anti-tumor immunity, human bladder cancer cell lines undergoing (dose-dependent apoptosis reaching approximately 30 % apoptotic cells using 150 μM CPS) apoptosis was shown to emit ICD hallmarks after treatment with CPS in vitro.

2) A figure detailing the pathway for CRT exposure would be helpful as well as better explanation of the role of ER stress and the UPR.  These are minimally discussed.

Thank you for pointing out the missing details, this issue has been discussed in the revised version of the manuscript. We designed a new figure(fig3) as you mentioned.

It is confirmed that ER stress is a significant factor in the ICD process. ICD inducers generate a form of ER stress that can be sensed by the immune cells through the release or surface exposure of DAMPs including CRT. The cellular response to ER stress leads to the stimulation of a signaling network called unfolded protein response (UPR). The stimulation of the UPR is correlated with three ER stress sensors: inositol-requiring enzyme 1 (IRE1), protein kinase R-like ER kinase (PERK), and activating transcription factor 6 (ATF6). In normal conditions, these proteins are maintained an inactive state by the master regulator of the UPR, GRP78/binding protein (BIP). While, when intra/extracellular stressors generate unfolded proteins in the ER, BiP separates from these ER stress sensors, thereby stimulating the UPR signaling network. During ICD, while other sensors of the UPR are activated, yet only the PERK is so far compulsive for ICD. PERK dimerizes, autophosphorylates, and in turn phosphorylates eIF2a that stops protein synthesis to reduce ER protein synthesis amount. CTR exposure in response to ICD inducers needs downstream of ER stress, caspase-8-based cleavage of the ER-resident protein, BAP31, and stimulation of Bax and Bak. ER-derived CRT is exocytosed through a classical, SNARE-dependent pathway. Exocytic vesicles originating from the Golgi have to fuse with the plasma membrane by molecular interactions between vesicle-related SNAREs and plasma membrane-related SNAREs ( SNAP23/25). Along all the way from the ER to the plasma membrane, CRT is accompanied by ERp57.This interaction is necessary for CRT translocation because the deletion of ERp57 stops CRT exposure. It is noteworthy to state that upon treatment with Type II ICD inducers fewer demands are required, since this pathway only depends on PERK, Bax, Bak, and the secretory pathway.

3) The title of the review does not really reflect the content since the discussion of the combination therapies is almost at the end and most of the review is general background on ICD. Recommending reducing this background content to what is relevant to introduce section 6.

Thank you for the mentioned point. During revision, a general evaluation was conducted. We have added more details in the main text about ICD and combination therapy approaches.

4) Other than section 6 - which is the novel and critical part of the review, the information presented is old (many references are decades old) and reported in other similar reviews.  In fact there are only a couple of references from 2018 and one from 2019.  So much of the material is dated.

We appreciate your comments. We did several modifications to the initial manuscript based on that your suggestions, in the revised version of the manuscript, we added new sections and updated most of the references. we hope the newer version shows more focus on the goal.

5) Addition of a table or figure for section 6 would bolster this information.  It is highly suggested to look at recent publications in 2019 and 2020 on ICD and combination therapy approaches, for example targeting the PD1-PD-L1 pathway to update the review.

We appreciate your suggestion for adding recent publications on ICD and combination therapy approaches. We added a new section as you mentioned. We summarized in a table defined as Table 2 in the main text.

After cessation of the chemotherapy process, residual tumor cells are able to upregulate adaptively the programmed death ligand-1 (PD-L1) so as to interact with the receptor of the immune checkpoint on the surface of T cells, which is a kind of escape system from immunosurveillance (immunosuppressive state). Therefore, there is a significant challenge in immunotherapy applications in cancers with immunosuppressive state [237]. It has been verified that monoclonal antibodies application for blocking of immune checkpoints could potentially provide high clinical achievement for cancer immunotherapy; however, their application is limited due to the cold environment of tumors [238]. Recently, many attentions have been focused to develop new and effective inhibitors for immune checkpoints in the immuno-oncology research area. Tumor cells could potentially escape from immunosurveillance and proceed with their specific mechanisms such as the pathway activation of immune checkpoints to repress the anti-tumor responses of the immune system. The inhibitors of immune checkpoints are able to revitalize the anti-tumor responses of the immune system through interfering with inhibitory effects on signaling pathways and also enhance the elimination of tumor cells by immune-mediated actions [239]. The implementation of such inhibitors against the PD-1/PD-L1 axis and probably CTLA-4 (cytotoxic T-lymphocyte antigen-4) has indicated considerable clinical benefits and even transformed rapidly as a practical method in medical oncology. Application of monoclonal antibodies, as potential co-inhibitors for immune checkpoints particularly CTLA-4 and PD-1, has been utilized clinically for various malignancies, including bladder cancer, head and neck squamous cell carcinoma, Hodgkin lymphoma, non-small cell lung cancer, Merkel cell carcinoma, melanoma, MSI-high colorectal carcinoma and renal cell carcinoma [240]. These inhibitors also have improved the possible practical methods for cancer treatments in the oncology field. Ipilimumab, as the first confirmed inhibitor for immune checkpoints that targets CTLA-4 molecules, has been applied for the treatment of those patients who suffer from advanced melanoma. This inhibitor is able to prevent the inhibition of T-cells and also promotes the proliferation and activation of T cells’ effectors. The other inhibitors, nivolumab, and pembrolizumab that target PD-1 indicated promising results in the treatment of NSCLC (non-small cell lung carcinoma) and melanoma with the rate of objective response between 40 and 45%. Furthermore, the treatment of urothelial bladder cancer with inhibitors of PD-1/PD-L1 pathways revealed an increment in response rate in the range of 13 and 24% [239].

Different clinical experiments have recently demonstrated that the use of anthracyclines as an ICD inducer led to an increase in the blocking of PD-1 and PD-L1 efficacy. These data could support the beneficial concept of ICD in cancer treatment [241]. LTX-401 is a known conjugated antibody- T-PNU drug, crizotinib as an inhibitor for tyrosine kinase and oxaliplatin as a chemotherapeutic agent, have a natural potential to induce ICD. It was reported that the combination of  LTX-401 and the inhibitors of CTLA-4 and PD-1  indicated considerable antineoplastic effects on primary distant and lesions tumors, showing the higher therapeutic effect of ICD and checkpoint blocker with together [241]. A study showed that lurbinectedin treatment could induce ICD traits such as calreticulin exposure, INF-I immune responses, ATP release and the exodus of HMGB1 molecules at in vitro condition.  The anti-cancer and antineoplastic properties of lurbinectedin could be strengthened remarkably by its combination with the double blockade of CTLA-4 and PD-1 immune checkpoints [242]. In another study conducted by Zao et al., it was revealed that irreversible electroporation (IRE) combined with anti-PD-1 could significantly induce the ICD process and also mediates the consequence durable response in a treated model (orthotopic PDAC). The mechanistic investigations disclosed that the efficiency of combined IRE plus anti-PD-1 is associated with several factors, including DCs activation, the prompt release of danger linked molecular patterns and alleviation of the immunosuppressive environment of tumor [243].

Tumor-treating fields (TTFields) are a well-known noninvasive anticancer treatment technique that applies alternating electric fields of intermediate frequency (∼100-500 kHz) with limited intensity (1-3 V/cm) to interrupt the division process of cancer cells [244]. TTFields therapy is practically implemented for the treatment of patients with malignant pleural mesothelioma and glioblastoma multiforme. It has been reported that TTFields therapy could potentially impair microtubules’ polymerization and filaments of septin, which both are needed mitosis procedure for proper cytokinesis and segregation of chromosomes. Therefore, the application of this method could lead to mitotic catastrophe and consequence cancer cells death associated with aneuploidy [245, 246]. Another study showed that TTFields therapy is able to induce the ICD process. The data of this study indicated that the combination of TTFields with anti-PD-1 for lung tumor treatment in the animal model led to a considerable reduction in tumor volume in comparison with the alone treatment or control group. Furthermore, TTFields combined anti-PD-1 therapy showed a remarkable increment in the infiltrating of CD45+ tumor cells [247]. PT-112, a new conjugate of platinum-pyrophosphate, is under development to be applied clinically for cancer treatment. In vitro studies reported that PT-112 could mediate the cytostatic and cytotoxic responses against numerous cell lines originated from human and mouse. The cytotoxic effects of PT-112 are related to the induction of danger signals initiating the anticancer immunity such as the exposure of calreticulin on the surface of dying cells and the release of HMGB1 and ATP molecules. It is demonstrated that the synergized administration of PT-112 and PD-1 or PD-L1 blockade in the animal model with immunologically competent settings could simultaneously induce immune effector cells and discharge immunosuppressive cells in the microenvironment of tumor [248].

HPMA copolymer with a degradable backbone has a significant advantage in targeting the facilitation of immunogenic drugs to improve their direct antitumor effects and ICD induction to provoke antitumor immunity. The copolymer, meanwhile, is able to mediate the crosslinking of PD-L1 on the surface according to its direction to the lysosome for degradation and also reveals persistent suppression. Furthermore, this two-pronged approach could employ and vivifies slumbering T cells in the microenvironment of tumors and provoke the responses of T cells. KT-1 and MPPA, as a new framework for the application of polymer-based nanomedicines, can be used to reprogram immune responses in “cold” tumors and also strengthen the immune system to attack cancer cells. KT-1 could considerably induce ICD in vivo condition and also sensitizes cancer cells to checkpoint blockade through targeting of polymer-enhanced tumors [237].  

Another study indicated that the encapsulated CRISPR/Cas9 in nanoparticles is able to knock out more specifically the function of cyclin-dependent kinase 5 gene to attenuate PD-L1 expression on cancer cells remarkably. More specifically, when paclitaxel (PTX) with a considerable capacity to convert “cold” tumor into “hot” tumor, encapsulated in noted nanoparticles could remarkably reduce regulatory T lymphocytes, enhance antitumor immunity, induce immunogenic cell death and repolarize tumor-associated macrophages [238].

Round 2

Reviewer 1 Report

The authors have been replied carefully to all the suggestions of the reviewers.

Reviewer 2 Report

The authors addressed all previous all concerns and improved the manuscript.